# CO$_2$ signaling mediates neurovascular coupling in the cerebral cortex

Patrick S. Hosford [1✉], Jack A. Wells [2], Shereen Nizari[1], Isabel N. Christie[1], Shefeeq M. Theparambil[1], Pablo A. Castro[3,4], Anna Hadjihambi[1], L. Felipe Barros [3], Iván Ruminot [3✉], Mark F. Lythgoe[2] & Alexander V. Gourine [1✉]

Neurovascular coupling is a fundamental brain mechanism that regulates local cerebral blood flow (CBF) in response to changes in neuronal activity. Functional imaging techniques are commonly used to record these changes in CBF as a proxy of neuronal activity to study the human brain. However, the mechanisms of neurovascular coupling remain incompletely understood. Here we show in experimental animal models (laboratory rats and mice) that the neuronal activity-dependent increases in local CBF in the somatosensory cortex are prevented by saturation of the CO$_2$-sensitive vasodilatory brain mechanism with surplus of exogenous CO$_2$ or disruption of brain CO$_2$/HCO$_3^-$ transport by genetic knockdown of electrogenic sodium-bicarbonate cotransporter 1 (NBCe1) expression in astrocytes. A systematic review of the literature data shows that CO$_2$ and increased neuronal activity recruit the same vasodilatory signaling pathways. These results and analysis suggest that CO$_2$ mediates signaling between neurons and the cerebral vasculature to regulate brain blood flow in accord with changes in the neuronal activity.

[1] Centre for Cardiovascular and Metabolic Neuroscience, Neuroscience, Physiology and Pharmacology, University College London, London, UK. [2] UCL Centre for Advanced Biomedical Imaging, Division of Medicine, University College London, London, UK. [3] Centro de Estudios Científicos (CECs) & Universidad San Sebastián, Valdivia, Chile. [4] Universidad Austral de Chile, Valdivia, Chile. ✉email: p.hosford@ucl.ac.uk; iruminot@cecs.cl; a.gourine@ucl.ac.uk

Neurovascular coupling is a fundamental, evolutionarily conserved signaling mechanism responsible for dilation of cerebral blood vessels and increase in local cerebral blood flow (CBF) in response to heightened neuronal activity. Supporting dynamic neuronal metabolic needs by maintaining an uninterrupted delivery of oxygen and glucose is thought to be the main purpose of the neurovascular coupling response. Functional imaging techniques, such as magnetic resonance imaging (fMRI), record these changes in local CBF as a proxy of neuronal activity to study the human brain. Recent evidence suggests that sustained impairment of neurovascular coupling may precipitate age-related neuronal damage, contribute to cognitive decline and the development of neurodegenerative disease[1,2]. Therefore, a full understanding of the signaling mechanisms between the brain neurons and the cerebral vasculature may prove to be important for the development of future treatments of these conditions, as well as for our understanding and interpretation of human brain imaging data.

However, there are still controversies surrounding the functional significance and the mechanisms underlying the neurovascular coupling response[3–5]. In accord with the currently prevailing view, neurovascular response is driven by neuronal activity-induced changes in the brain neurochemical milieu (neurotransmitter spillover, increase in extracellular $K^+$) directly leading to vascular responses[6] and/or activation of intermediate cell types, including interneurons and astrocytes, which in turn signal to vascular smooth muscle cells and pericytes[7]. A recent systematic review and meta-analysis of published data obtained in studies involving experimental animal models and reporting the effects of pharmacological or genetic blockade of all hypothesized signaling pathways indicated that such feed-forward mechanisms may account for up to ~60% of the neurovascular coupling response[5]. The analysis pointed to the existence of an as yet unidentified signaling mechanism(s) responsible for a significant proportion (at least one third) of the response[5].

In air-breathing animals with a high metabolic rate living in conditions of ample oxygen supply, the effective removal of metabolically produced $CO_2$ is critical to maintain homeostasis. The human brain generates ~20% of total body $CO_2$ production (3.3 moles or ~75 l per day) and this $CO_2$ can only be removed from the brain by the cerebral circulation. All membrane, molecular and biochemical processes involved in synaptic transmission are affected by changes in pH, therefore, uncontrolled fluctuations in brain tissue $CO_2$/pH are detrimental to neuronal function. $CO_2$ production increases in parallel with neuronal activity and energy usage[8] and has a very potent dilatory effect on the brain vasculature[9], which (in contrast to systemic vessels) is uniquely sensitive to $CO_2$[10]. Conceivably, the need for effective removal of surplus $CO_2$ generated during increased brain activity and brain pH regulation was an important driving force behind the evolutionary development of the neurovascular coupling mechanism. If so, $CO_2$ may be expected to act as a signaling molecule between active neurons and brain vessels. Yet, rather surprisingly, the role of locally produced $CO_2$ in the mechanisms underlying the neurovascular coupling response has never been experimentally addressed.

Here we show in experimental animal models (rats and mice) that the neuronal activity-dependent increases in local CBF in the somatosensory cortex are prevented by saturation of brain $CO_2$-sensitive vasodilatory mechanism with surplus of exogenous $CO_2$ or disruption of brain $CO_2$/$HCO_3^-$ transport by conditional deletion of electrogenic sodium-bicarbonate cotransporter 1 (NBCe1) in astrocytes. A systematic literature review of the data obtained in humans and experimental animals on the mechanisms underlying cerebrovascular response to $CO_2$ show that $CO_2$ and increased neuronal activity recruit the same vasodilatory

signaling pathways. These results and analysis suggest that $CO_2$ signaling mediates the neurovascular coupling in the cerebral cortex.

## Results

**Signaling mechanisms underlying the cerebrovascular response to $CO_2$: a systematic review of literature data**. To test the hypothesis that $CO_2$ mediates the neurovascular coupling response, methods of specific blockade of $CO_2$ transport, sensing or actions in the brain are required. If our hypothesis is correct, then the blockade of signaling mechanisms that mediate $CO_2$-induced cerebrovascular dilations should also block the development of the neurovascular response. Literature searches return hundreds of studies describing multiple signaling pathways that can potentially mediate the effects of $CO_2$ on brain perfusion, yet according to a recent review the exact mechanisms underlying cerebrovascular responses to $CO_2$ remain unclear[9]. To identify potential target(s) for experimental manipulation, we first evaluated the relative significance of different signaling mechanisms of cerebrovascular $CO_2$ sensitivity, suggested by the preceding studies. We conducted a systematic review and meta-analysis of published data obtained in the experimental animal and human studies that targeted all hypothesized mechanisms either pharmacologically or genetically. Our primary outcome measure was the percent reduction of the cerebrovascular response to $CO_2$, recorded in in vivo animal models and in studies involving human participants.

Selection criteria were met by 131 primary sources (Supplementary References), reporting the data obtained under 214 different experimental conditions (Fig. 1 and Supplementary Table 1). Experimental conditions were grouped into categories that included studies targeting mechanisms mediated by nitric oxide (NO), cyclooxygenase products, adenosine, amongst others. The largest number of published studies targeted cyclooxygenase and NO-mediated pathways. Inhibition of cyclooxygenase with indomethacin, genetic, or pharmacological blockade of the neuronal NO synthase (NOS), and non-specific NOS inhibition were found to have comparable effects in reducing the cerebrovascular response to $CO_2$ by 44% (average of 30 experimental animal and 18 human studies), 40% (average of 18 animal studies), and 39% (average of 54 animal and 3 human studies), respectively (Fig. 1). Blockade of cyclooxygenase-1 was found to have the largest effect of inhibiting any specific individual target, reducing the $CO_2$-induced cerebrovascular response by 56% (average of 4 animal studies). It is important to note that non-specific cyclooxygenase inhibitors had on average a substantially smaller effect (inhibition by 7%; average of 4 experimental animal and 8 human studies), compared to that of indomethacin or specific cyclooxygenase-1 blockade (Fig. 1). Of all the experimental conditions, combined treatment with indomethacin and NOS blockade was identified as the most efficacious in inhibiting the cerebrovascular response to $CO_2$ (inhibition by 87%; average of 6 experimental animal studies) (Fig. 1).

This analysis showed that $CO_2$ actions in the brain recruit the same vasodilatory signaling pathways, involving nitric oxide and cyclooxygenase products, that are implicated in mediating responses of cerebral blood vessels to the increases in the neuronal activity[5,7]. Although, the outcome of this analysis supported the main hypothesis of this study, it was not useful in terms of informing the next experiments, as all the key identified signaling pathways were targeted in many previous studies of the neurovascular coupling response[5]. Moreover, while the vasodilatory signaling pathways that mediate the $CO_2$ effects had been extensively researched, the cellular and molecular mechanisms underlying cerebrovascular $CO_2$ sensing are not well understood. Therefore, conventional pharmacological or genetic approaches cannot be applied with certainty to block these mechanisms.

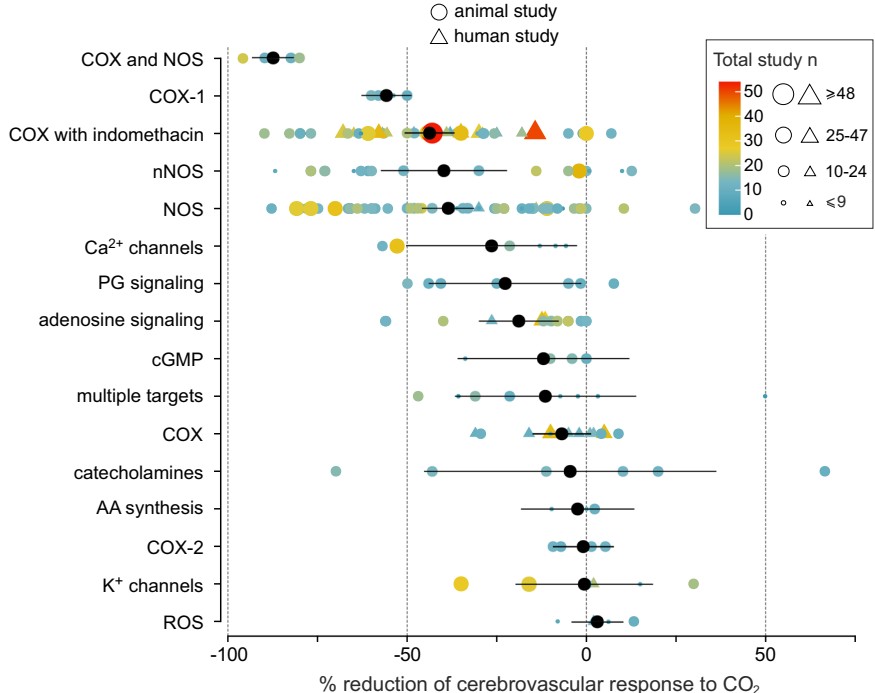

**Fig. 1 Summary data illustrating the outcome of systematic review and analysis of data reported in studies that investigated the signaling mechanisms underlying the cerebrovascular response to CO₂.** Individual data points illustrate the percent reduction of the cerebrovascular response to $CO_2$ in conditions of pharmacological or genetic blockade of hypothesized signaling pathways, reported in published studies involving in vivo animal models and human participants. Color gradient and the size of each symbol denote the statistical power of each individual study (total study *n*). Black symbols illustrate the mean (with 95% confidence intervals) reduction of $CO_2$-induced cerebrovascular response observed under experimental conditions (214 in total, reported in 131 primary sources; Supplementary Table 1) grouped into categories of studies targeting specific signaling pathways. The effects of indomethacin are analyzed and shown in a separate category as inhibition of cerebrovascular $CO_2$ reactivity by this drug is greater compared to that of other non-specific cyclooxygenase inhibitors. Multiple category includes the results of experimental studies that targeted several signaling pathways simultaneously. COX cyclooxygenase, nNOS neuronal nitric oxide synthase, NOS nitric oxide synthase, PG prostaglandin, cGMP cyclic guanosine monophosphate, AA arachidonic acid, ROS reactive oxygen species. Source data are provided as a Source Data file.

**CO₂ added to the inspired air blocks the neurovascular response in the cerebral cortex of rats**. To study the role of $CO_2$-mediated signaling in the development of the neurovascular response, we used an alternative experimental strategy which involved saturation of the brain $CO_2$-sensitive vasodilatory mechanism(s) with exogenous $CO_2$ given in the inspired air—a condition referred herein as hypercapnia (Fig. 2a). We reasoned that if $CO_2$ acts as a signaling molecule between the brain neurons and the vasculature then in conditions of surplus of exogenous $CO_2$ any additional metabolic $CO_2$ generated as a result of increased neuronal activity should have no further effect on the cerebral vessels and the neurovascular response should be blocked (Fig. 2a). However, if mechanisms other than $CO_2$ signaling mediate the neurovascular response, then its development should not at all be affected by hypercapnia.

In the experiments described next, we used laboratory rats and implemented an arterial spin labeling (ASL) fMRI sequence with T2* weighted imaging for combined measurement of local CBF (in ml $100 \, g^{-1} \, min^{-1}$) and blood oxygen level dependent (BOLD) signal changes in the forelimb region 1 of the somatosensory cortex (S1FL) (Fig. 2b). Under normocapnic conditions (arterial PCO₂ ~35 mmHg) activation of somatosensory pathways by electrical forepaw stimulation triggered robust CBF increases (from $88 \pm 10$ to $120 \pm 11$ ml $100 \, g^{-1} \, min^{-1}$, $p = 0.015$, $n = 7$) and BOLD responses ($1.3 \pm 0.5\%$, $p = 0.008$, $n = 7$) in the S1FL region (Fig. 2c). The addition of 5% and 10% inspired $CO_2$ increased global CBF (Fig. 2c, d) and reduced the magnitude of the CBF responses to somatosensory stimulation by 55% ($p = 0.016$) and 87% ($p = 0.006$), respectively (Fig. 2c, e).

The BOLD response was reduced by 83% ($0.33 \pm 0.2\%$, $p = 0.013$) in conditions of 10% inspired $CO_2$ (Fig. 2c, f).

**Surplus of exogenous CO₂ blocks the neurovascular response independently of changes in brain tissue pH**. Next, we determined whether the loss of the neurovascular response during hypercapnia is due to acidification of the brain tissue resulting from $CO_2$ hydration. We manipulated the brain extracellular pH by systemic administration of a carbonic anhydrase inhibitor acetazolamide ($10 \, mg \, kg^{-1}$, i.v.), which is well known to reduce pH within the brain. Acetazolamide decreased brain tissue pH by the same degree as 5% inspired $CO_2$ (decrease by $0.11 \pm 0.01$ pH units after acetazolamide vs. $-0.10 \pm 0.01$ pH units in conditions of 5% inspired $CO_2$, $p > 0.9$; Fig. 2g). Acetazolamide increased the basal CBF but had no effect on cortical neuronal, CBF and BOLD responses induced by somatosensory stimulation (Fig. 2c, e, f and Supplementary Figs. 1 and 3). Thus, in conditions of a similar degree brain tissue acidification, the neurovascular response in the S1FL region induced by forepaw stimulation was strongly inhibited by 5% inspired $CO_2$, but was unaffected following systemic acetazolamide treatment (Fig. 2c, e, f and Supplementary Fig. 1). No effect of acetazolamide on the expression of the neurovascular response was previously observed in human studies[11].

Interestingly, acetazolamide also had no effect on the magnitude and time-course of CBF increases induced by 5% and 10% inspired $CO_2$ (Supplementary Fig. 2). Thus, both the increases in local CBF induced by heightened neuronal activity

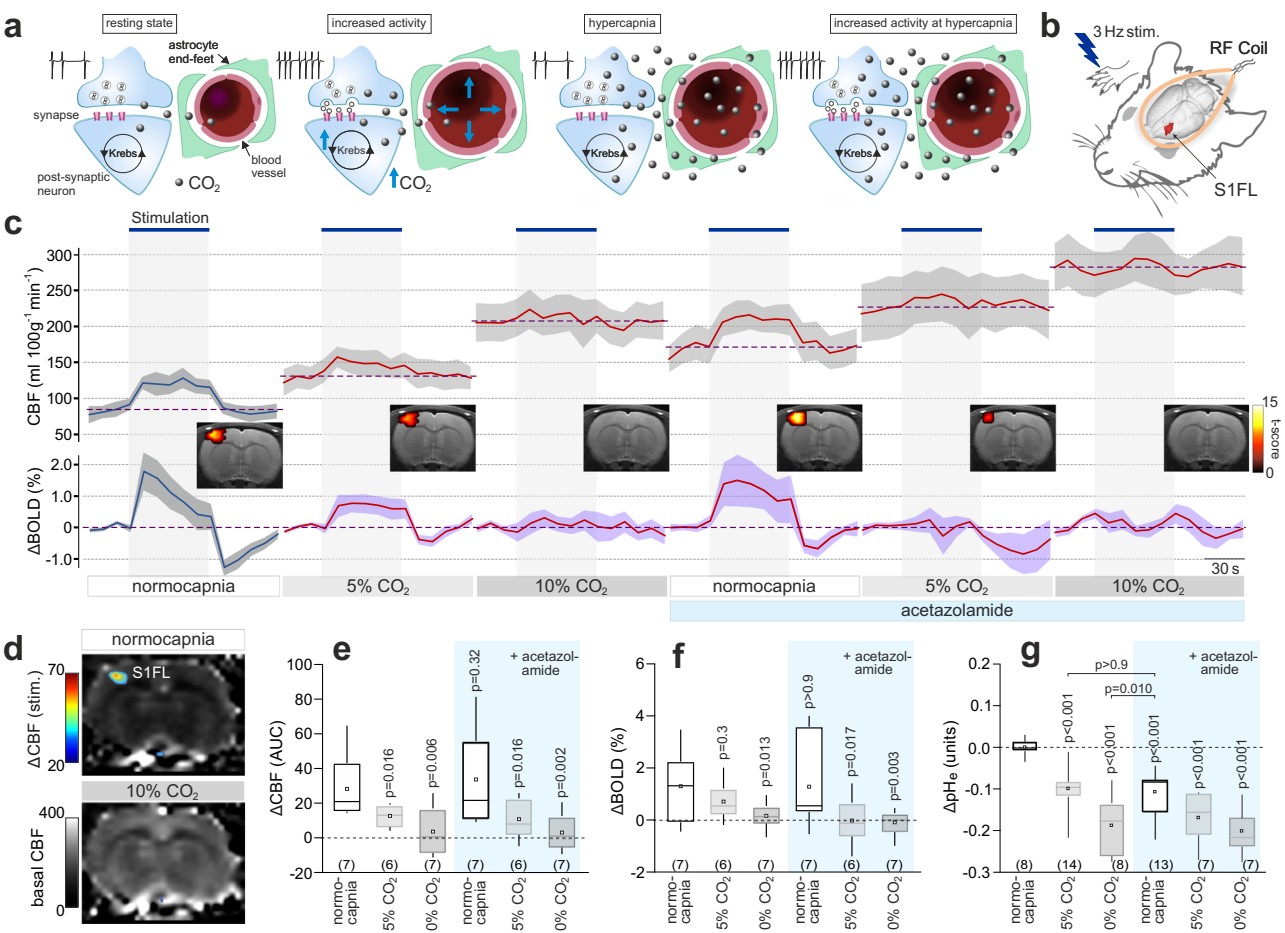

**Fig. 2 Exogenous $CO_2$ prevents the development of the neurovascular response in the somatosensory cortex in rats. a** Schematic depiction of the neurovascular unit illustrating the central hypothesis and the experimental approach taken to study the potential role of $CO_2$ as a signaling molecule that mediates the neurovascular coupling response. A surplus of exogenous $CO_2$ was given in the inspired air (hypercapnia) in order to saturate the brain $CO_2$-sensitive vasodilatory mechanism. If the hypothesis is correct, then in the presence of surplus of exogenous $CO_2$, any extra metabolic $CO_2$ generated as a result of increased neuronal activity should have no additional effect on local cerebral blood flow (CBF) and the neurovascular response should be blocked. If the mechanisms other than $CO_2$ signaling mediate the neurovascular coupling response, then the response should not be affected by the excess of exogenous $CO_2$. **b** CBF and blood oxygen level dependent (BOLD) signals in the forepaw region of the somatosensory cortex (S1FL) in anesthetized rats were recorded using an arterial spin labeling (ASL) sequence with T2* weighted imaging. Somatosensory pathways were activated by electrical stimulation of the forepaw. RF radiofrequency. **c** CBF and BOLD responses in the S1FL region induced by electrical forepaw stimulation (3 Hz, 1.5 mA) at baseline, in conditions of 5% and 10% inspired $CO_2$, after the administration of carbonic anhydrase inhibitor acetazolamide (10 mg kg$^{-1}$, i.v.), and in conditions of 5% and 10% inspired $CO_2$, applied concomitantly with systemic carbonic anhydrase inhibition with acetazolamide. Data are presented as mean values ± SEM (shaded areas denote error bands). *Insets* show representative activation maps illustrating mean BOLD signal changes in response to somatosensory stimulation. Color bar: *t*-score from statistical paramagnetic mapping mixed-effects analysis, $p < 0.05$ (uncorrected). **d** Representative ASL images illustrating CBF at rest (normocapnia) and in conditions of 10% inspired $CO_2$. Overlaid (false color scale) illustrates CBF response in the S1FL region induced by forepaw stimulation. **e, f** Summary data illustrating the effect of 5% and 10% inspired $CO_2$, given before and after systemic administration of acetazolamide, on CBF and BOLD responses recoded in the S1FL region of the cortex. *P* values, mixed-model ANOVA followed by Holm–Sidak multiple comparison test. **g** Summary data illustrating peak changes in brain tissue pH (S1FL cortical region) following administration of acetazolamide, as well as in response to 5% and 10% inspired $CO_2$, given before and after systemic administration of acetazolamide. *P* values, Kruskal–Wallis test followed by Dunn's multiple comparison test. In the box-and-whisker plots the central dot indicates the mean, the central line indicates the median, the box limits indicate the upper and lower quartiles, and the whiskers show the minimum–maximum range of the data. Numbers in parentheses indicate sample sizes (number of animals per experimental group). Source data are provided as a Source Data file.

and increases in global CBF induced by inhalation of $CO_2$ were unaffected in conditions of brain tissue acidification induced by acetazolamide. These data suggest that carbonic anhydrase activity plays no significant role in the mechanisms underlying the effects of $CO_2$ on cerebral vasculature. $CO_2$ and $H_2O$ are converted to $HCO_3^-$ and $H^+$ without the involvement of carbonic anhydrase at a relatively high speed with a first-order rate constant of 0.15 s$^{-1}$ (10$^6$ s$^{-1}$ with carbonic anhydrase II)[12]. If the mechanisms of cerebrovascular $CO_2$ sensitivity involve $CO_2$

hydration, then the rate of non-enzymatic conversion appears to be sufficient for these mechanisms to operate in conditions when carbonic anhydrase is inhibited by acetazolamide.

Systemic treatment with acetazolamide was however insufficient to fully mimic the effect of 10% inspired $CO_2$ on brain tissue pH (decrease by 0.11 ± 0.01 pH units after acetazolamide vs. 0.19 ± 0.02 pH units at 10% inspired $CO_2$, $p = 0.010$; Fig. 2g). Therefore, it could be argued that in conditions of 10% inspired $CO_2$, almost complete blockade of the neurovascular response can

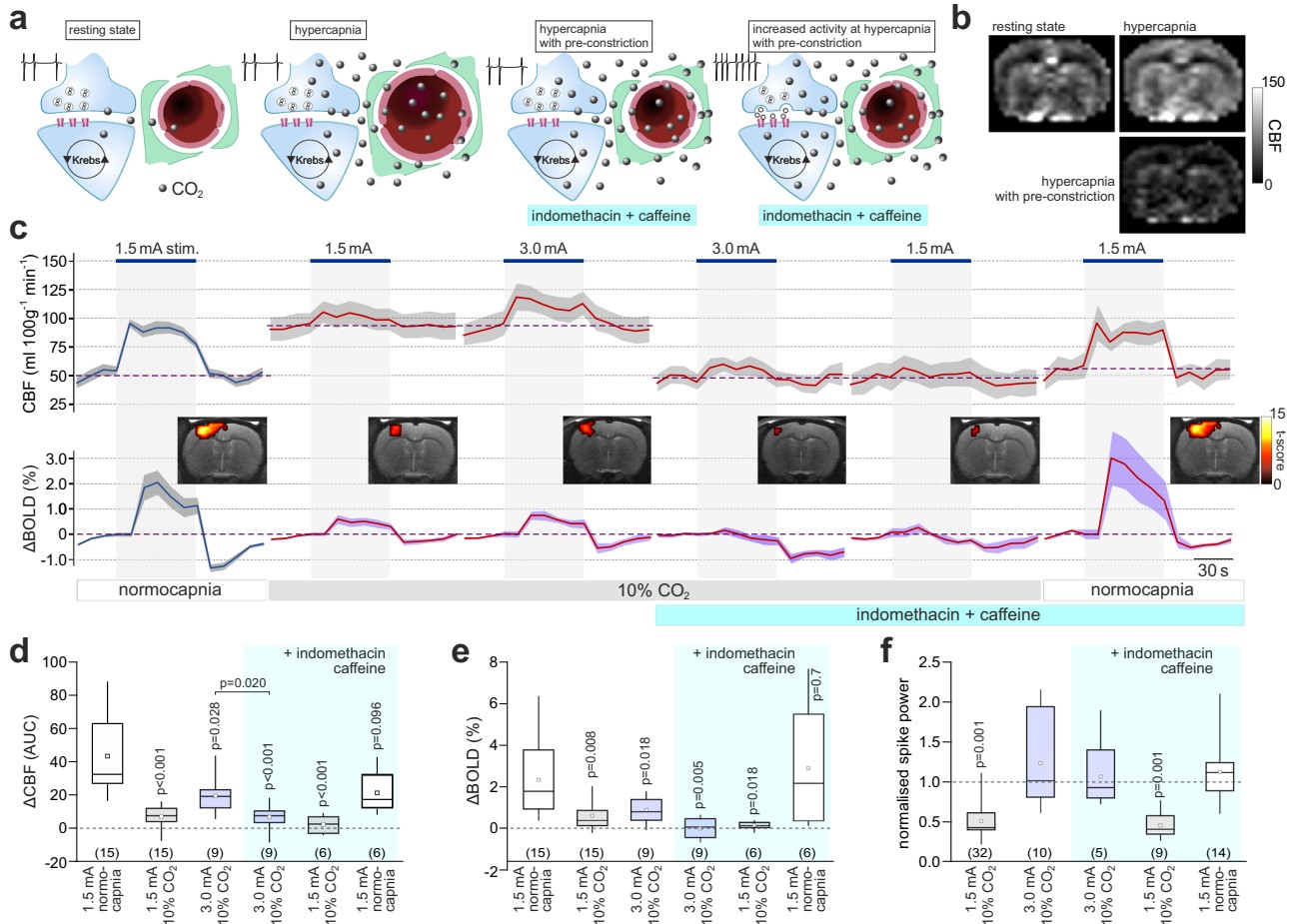

**Fig. 3 Exogenous CO₂ prevents the development of the neurovascular response in the somatosensory cortex independently of changes in basal CBF.**
**a** Schematic depiction of the neurovascular unit illustrating the experimental approach used to control for the effect of systemic hypercapnia on basal CBF. Extra $CO_2$ generated during enhanced neuronal activity has no effect on local blood flow in the presence of surplus of exogenous $CO_2$. However, under these conditions the cerebral vasculature is dilated and may lack the capacity to dilate further. To counteract the action of 10% inspired $CO_2$ on brain vasculature, the animals were given a combination of caffeine and indomethacin (both at 10 mg kg$^{-1}$, i.v.) to reduce the CBF to the baseline level recorded at normocapnia. **b** Representative ASL images illustrating CBF at rest (normocapnia), in conditions of 10% inspired $CO_2$, and after the systemic treatment with caffeine/indomethacin at 10% inspired $CO_2$. **c** CBF and BOLD responses in the S1FL region induced by forepaw stimulation at baseline, in conditions of 10% inspired $CO_2$, after the systemic administration of caffeine and indomethacin in conditions of 10% inspired $CO_2$, and after the withdrawal of inspired $CO_2$. To compensate for $CO_2$-induced inhibition of the neuronal activity, the intensity of forepaw stimulation was increased two-fold from 1.5 to 3 mA when 10% inspired $CO_2$ was applied. Data are presented as mean values ± SEM (shaded areas denote error bands). *Insets* show representative activation maps illustrating mean BOLD signal changes in response to somatosensory stimulation. Color bar: *t*-score from statistical paramagnetic mapping mixed-effects analysis, $p < 0.05$ (uncorrected). **d**, **e** Summary data illustrating CBF and BOLD responses in the S1FL region induced by electrical forepaw stimulation at baseline, in conditions of 10% inspired $CO_2$, evoked by 3 mA stimulus intensity in conditions of 10% inspired $CO_2$, after the systemic administration of caffeine/indomethacin in conditions of 10% inspired $CO_2$, and after the withdrawal of inspired $CO_2$. *P* values, Kruskal–Wallis test followed by Dunn's multiple comparison test. **f** Summary data illustrating the effect of each of the experimental conditions on the neuronal responses (expressed as spike power) in the S1FL region triggered by electrical forepaw stimulation. Integrated spike activity was normalized and presented relative to the responses recorded at baseline. *P* values, one-way ANOVA followed by Tukey's multiple comparison test. In the box-and-whisker plots the central dot indicates the mean, the central line indicates the median, the box limits indicate the upper and lower quartiles, and the whiskers show the minimum–maximum range of the data. Numbers in parentheses indicate sample sizes (number of animals per experimental group). Source data are provided as a Source Data file.

be explained by the inhibitory effect of low pH on the synaptic/ neuronal activity, and not by the saturation of the $CO_2$-sensitive vasodilatory mechanism with exogenous $CO_2$, as intended by the experimental design.

Analysis of the evoked neuronal activity in the S1FL region revealed the inhibitory effect of $CO_2$. The S1FL neuronal responses to somatosensory stimulation were reduced by 34% ($p = 0.04$) and 48% ($p = 0.001$) in conditions of 5% and 10% inspired $CO_2$, respectively (Fig. 3f and Supplementary Fig. 3). Interestingly, acetazolamide had no effect ($p > 0.9$) on the evoked neuronal activity (Supplementary Fig. 3), suggesting that the

inhibitory effect of $CO_2$ is independent of the associated changes in brain pH. Increasing the intensity of forepaw stimulation two-fold, from 1.5 to 3 mA, fully compensated (and exceeded by 25% the responses evoked by 1.5 mA stimulation) for the $CO_2$-induced inhibition (Fig. 3f and Supplementary Fig. 3), but failed to restore the neurovascular response in the S1FL region, with both the CBF and BOLD responses remained markedly suppressed by 10% inspired $CO_2$ (by 54%, $p = 0.020$ and 61%, $p = 0.018$, respectively; Fig. 3c–e). Thus, surplus of exogenous $CO_2$ still effectively inhibited the neurovascular response in the somatosensory cortex triggered by forepaw stimulation, which was applied with

increased intensity in order to offset the inhibitory effect of $CO_2$ on the neuronal activity.

Collectively these data support the conclusion that $CO_2$ added to the inspired air inhibits, and in higher concentration blocks, the neurovascular response by saturation of the $CO_2$-sensitive vasodilatory mechanism (effectively blocking further activation), independently of changes in brain tissue pH and the effect of $CO_2$ on the neuronal activity.

**Exogenous $CO_2$ blocks the neurovascular response independently of changes in brain perfusion.** As expected, $CO_2$ added to the inspired air increased the global CBF (Figs. 2c, d; 3b, c; Supplementary Fig. 4). Therefore, it could be argued that during hypercapnia increased neuronal activity fails to trigger a neurovascular response over and above the elevated CBF because the cerebrovascular reserve is exhausted by the potent dilatory action of $CO_2$ on brain vasculature. In conditions of systemic acetazolamide action, 10% inspired $CO_2$ increased the CBF to ~280 ml $100\,g^{-1}\,min^{-1}$ (Fig. 2c), likely revealing the cerebrovascular reserve in our experimental model. $CO_2$ given in the inspired air in concentrations of 5% and 10% increased the CBF to ~130 and 210 ml $100\,g^{-1}\,min^{-1}$, respectively, while the peak CBF response to somatosensory stimulation (i.e. the difference between the baseline CBF and peak increase in CBF) recorded in control conditions (normocapnia) was $39 \pm 5$ ml $100\,g^{-1}\,min^{-1}$ ($n = 22$) (Figs. 2c, e; 3c, d). As the sum of these CBF values (210 ml $100\,g^{-1}\,min^{-1}$ peak CBF at 10% $CO_2$ + 39 ml $100\,g^{-1}\,min^{-1}$ peak neurovascular response) is lower than the CBF recorded at hypercapnia following systemic acetazolamide treatment (280 ml $100\,g^{-1}\,min^{-1}$), the inhibition of the neurovascular response by exogenous $CO_2$ cannot simply be explained by the exhaustion of the cerebrovascular reserve.

In the next series of experiments to counteract the effects of exogenous $CO_2$ and lower the brain perfusion to the baseline level recorded in normocapnia, we gave the animals 10% $CO_2$ in the inspired air and applied pharmacological agents known to reduce the CBF—caffeine[13] and indomethacin[14] (Fig. 3a–c and Supplementary Figs. 4 and 5). The actions of neither caffeine (10 mg kg⁻¹, i.v.) nor indomethacin (10 mg kg⁻¹, i.v.) given alone were able to fully offset the effect of inspired $CO_2$ and reduce the CBF to the baseline level (Supplementary Fig. 5). However, combined treatment with caffeine and indomethacin during hypercapnia effectively restored the baseline (normocapnic) level of CBF ($49 \pm 6$ ml $100\,g^{-1}\,min^{-1}$ in 10% inspired $CO_2$ following caffeine/indomethacin administration vs. $53 \pm 4$ ml $100\,g^{-1}\,min^{-1}$ at baseline; $p = 0.77$) (Fig. 3b, c and Supplementary Fig. 4). Although in conditions of 10% inspired $CO_2$ the CBF was lowered to the baseline by this treatment, no significant CBF and BOLD responses to somatosensory stimulation were recorded in the S1FL region (reduction by 83%, $p < 0.001$ and 94%, $p = 0.005$; respectively; Fig. 3c–e).

In these experiments, forepaw stimulations were applied at two different intensities, 1.5 and 3 mA. As described above, increasing the intensity of stimulation from 1.5 to 3 mA was required in order to fully compensate for the $CO_2$-induced inhibition of the neuronal activity in the S1FL region (Fig. 3f). Increasing the intensity of forepaw stimulation led to a partial recovery of the CBF response in conditions of 10% inspired $CO_2$ (to 46% of the control response). As no significant CBF and BOLD responses to somatosensory stimulation were recorded at 10% inspired $CO_2$ following systemic treatment with caffeine/indomethacin (Fig. 3c–e), this pharmacological treatment appeared to have some inhibitory effect on the neurovascular response in high $CO_2$ conditions; the CBF response in the S1FL region induced by 3 mA stimulation at 10% inspired $CO_2$ was reduced by 29% ($p = 0.020$) following systemic treatment with caffeine and indomethacin (Fig. 3d). This result is consistent with the data reported in the literature[5], suggesting that the

signaling pathways sensitive to blockade by caffeine and indomethacin partially contribute to the development of the neurovascular response. However, these data also show that the bulk of the response is mediated by a separate (COX- and adenosine receptor-independent) mechanism, operation of which is fully blocked by a surplus of exogenous $CO_2$.

The neurovascular response to somatosensory stimulation rapidly recovered after the withdrawal of $CO_2$ from the inspired air; there were no differences in the neuronal, CBF and BOLD responses in the S1FL triggered by 1.5 mA forepaw stimulation at baseline and in conditions of systemic caffeine and indomethacin action (Fig. 3c–f). The recovery of the neurovascular response after the $CO_2$ withdrawal cannot be explained by the washout of caffeine and indomethacin during the course of the experiment (<90 min), as the half-life of both drugs in rodents is >3 h[15].

**Disruption of brain $CO_2$ transport blocks the neurovascular response in the cerebral cortex of mice.** All penetrating and intraparenchymal cerebral blood vessels are wrapped by end-feet of astrocytes (Fig. 4a). To be removed by cerebral circulation, $CO_2$ generated by brain neurons, crosses at least two astroglial membranes to reach the cerebral vessels. Historically, $CO_2$ was thought to diffuse through the plasma membrane freely. More recent evidence led to the understanding that the diffusion of $CO_2$ across the biological membrane is restricted and, to a large extent, is mediated by certain membrane transporters and channels, such as aquaporin water channels[16]. One of the main proposed conduits of facilitated $CO_2$ transport across the membrane is electrogenic sodium-bicarbonate cotransporter 1 (NBCe1) which, in the presence of $HCO_3^-$, has a substantial $CO_2$ conductance[16]. NBCe1 is characteristically expressed by brain astrocytes[17–19], promotes fast $CO_2$/$HCO_3^-$ transport and helps to maintain brain pH homeostasis[19].

We next hypothesized that if $CO_2$ mediates signaling between brain neurons and cerebral vasculature then the development of the neurovascular coupling response should be affected if $CO_2$/$HCO_3^-$ transport is disrupted and $CO_2$/pH homeostasis is not effectively maintained (Fig. 4a). This hypothesis was tested in the experiments using mice with astrocyte-specific conditional genetic knockdown of NBCe1[19]. In this experimental animal model, NBCe1 expression in cortical astrocytes is reduced by ~40%[19] and displays a characteristic mosaic pattern (Fig. 4c). However, NBCe1 expression in astrocytes of the brainstem was unaffected (Supplementary Fig. 6), likely due to a high expression of NBCe1 in this area of the brain[20]. The resting respiratory activity, and $CO_2$-induced increases in ventilation were found to be normal (Supplementary Fig. 4), indicative of a normal global brain perfusion in this animal model. In the open field test, the animals with conditional NBCe1 knockdown in astrocytes displayed behavioral pattern of activity consistent with increased anxiety (Fig. 4d, e). As $CO_2$ is a potent trigger of anxiety behavior[21], these data support the hypothesis that NBCe1 deficiency impairs the cellular mechanisms of brain $CO_2$ clearance.

Using a minimally invasive method of fast cyclic voltammetry[22] to measure changes in brain tissue $PO_2$ (Fig. 4b), we recorded stereotypical neurovascular responses to somatosensory stimulation in the S1FL cortical region of mice. In the control animals (NBCe1$^{flox/flox}$:GLAST$^{CreERT2/+}$ mice treated with the vehicle), significant increases in brain tissue $PO_2$ (peak $3.7 \pm 1.7$ mmHg; $n = 7$; $p = 0.038$) were recorded at the end of the 20 s stimulation period, returning to the baseline within 70 s after the end of the stimulation (Fig. 4f). In conditions of NBCe1 knockdown in astrocytes (NBCe1$^{flox/flox}$:GLAST$^{CreERT2/+}$ mice treated with tamoxifen), the neurovascular response in the S1FL region induced by forepaw stimulation was completely abolished, with tissue $PO_2$ falling below the baseline during the period of stimulation (by

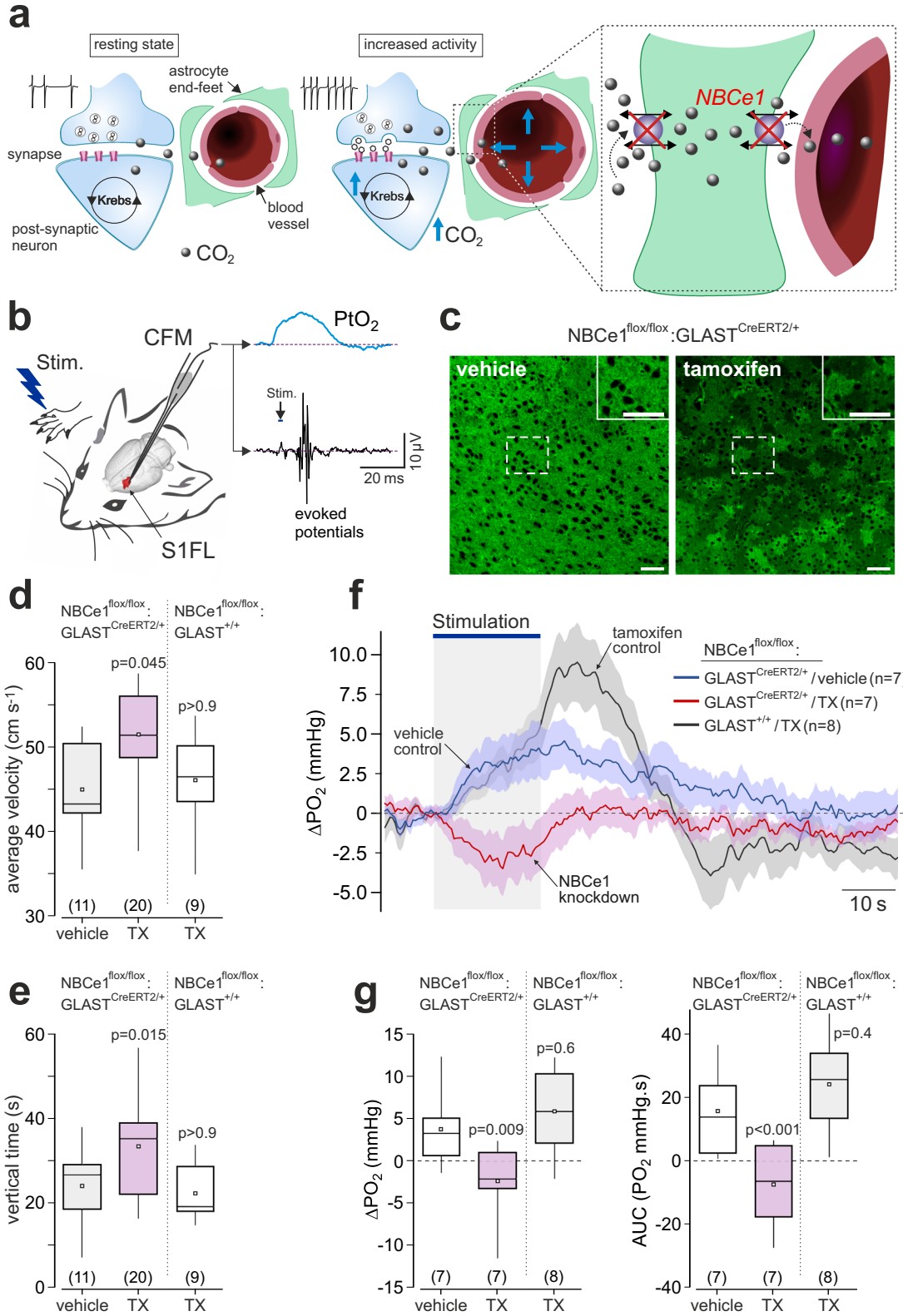

-2.4 ± 1.7 mmHg; n = 7), resulting in a marked difference in the integral PO₂ response between the experimental and two control groups (-7.4 ± 4.7 AUC in NBCe1-deficient mice, n = 7; vs. 15.8 ± 4.8 AUC in NBCe1^{flox/flox}:GLAST^{CreERT2/+} mice treated with the vehicle, n = 7; vs. 24.3 ± 5.0 AUC in NBCe1^{flox/flox}:GLAST^{+/+} mice treated with tamoxifen, n = 8; p = 0.009 and p < 0.001, respectively) (Fig. 4f, g).

## Discussion

Over 130 years ago Roy and Sherrington proposed that the "chemical products of cerebral metabolism … can cause variations of the caliber of the cerebral vessels: that in this reaction the brain possesses an intrinsic mechanism by which its vascular supply can be varied locally in correspondence with local variations of functional activity"[23]. To the best of our knowledge our study provides

**Fig. 4 Genetic knockdown of sodium bicarbonate cotransporter 1 (NBCe1) in astrocytes blocks the neurovascular response in the somatosensory cortex in mice. a** Schematic depiction of the neurovascular unit illustrating disrupted $CO_2/HCO_3^-$ transport in astroglial endfeet in conditions of NBCe1 deficiency. **b** Changes in tissue $PO_2$ ($PtO_2$) and the neuronal activity in the S1FL cortical region of mice were recorded using fast scan cyclic voltammetry. The carbon fiber microelectrode (CFM) was placed in the S1FL region. The forepaw was stimulated electrically to activate the somatosensory pathways. **c** Knockdown of NBCe1 expression in cortical astrocytes. Representative confocal images illustrating immunohistochemical detection of NBCe1 in the cortex of NBCe1$^{flox/flox}$:GLAST$^{CreERT2/+}$ mice treated with the vehicle (oil) or tamoxifen. Tamoxifen treatment of NBCe1$^{flox/flox}$:GLAST$^{CreERT2/+}$ mice resulted in a mosaic pattern of NBCe1 expression. Staining was repeated in four animals from each experimental group with similar result. Scale bars = 50 μm. **d, e** Activity in the open field exhibited by NBCe1$^{flox/flox}$:GLAST$^{CreERT2/+}$ mice treated with vehicle, NBCe1$^{flox/flox}$:GLAST$^{+/+}$ mice (cre-negative) treated with tamoxifen, and NBCe1$^{flox/flox}$:GLAST$^{CreERT2/+}$ mice treated with tamoxifen. Mice with conditional NBCe1 knockdown in astrocytes displayed behavioral pattern (increased locomotor speed and vertical activity) consistent with the increased anxiety. **f** Time course of $PtO_2$ changes in the S1FL region induced by activation of somatosensory pathways (electrical stimulation of the contralateral paw; 3 Hz, 1.5 mA) in NBCe1$^{flox/flox}$:GLAST$^{CreERT2/+}$ mice treated with the vehicle or tamoxifen and NBCe1$^{flox/flox}$:GLAST$^{+/+}$ mice treated with tamoxifen. NBCe1 knockdown in cortical astrocytes prevented the development of the neurovascular response. Traces illustrate averaged (mean values ± SEM) changes in $PO_2$-sensitive current. **g** Summary data illustrating peak (measurements taken at the end of the stimulation period) and integral (area under the curve, AUC) changes in $PO_2$, evoked in the S1FL region by electrical forepaw stimulation in NBCe1$^{flox/flox}$:GLAST$^{CreERT2/+}$ mice treated with the vehicle or tamoxifen, and NBCe1$^{flox/flox}$:GLAST$^{+/+}$ mice treated with tamoxifen. In the box-and-whisker plots the central dot indicates the mean, the central line indicates the median, the box limits indicate the upper and lower quartiles, and the whiskers show the minimum–maximum range of the data. Numbers in parentheses indicate sample sizes (number of animals per experimental group). $P$ values, Kruskal–Wallis test followed by Dunn's multiple comparison test. Source data are provided as a Source Data file.

the first experimental data in support of this oldest, but largely discounted, hypothesis of the mechanism that links neuronal activity with regulation of local cerebral blood flow via the actions of $CO_2$. The data obtained show that the neurovascular response in the cerebral cortex can be inhibited or completely blocked by experimental treatments involving saturation of the brain $CO_2$-sensitive vasodilatory mechanism by the provision of exogenous $CO_2$, or disruption of $CO_2/HCO_3^-$ transport and pH regulation following conditional knockdown of NBCe1 expression in astrocytes. Experiments involving addition of $CO_2$ to the inspired air in order to occlude the actions of metabolically produced (endogenous) $CO_2$ were carefully designed to control for $CO_2$-induced changes in brain tissue pH, neuronal activity, and perfusion. Collectively the data obtained in these experiments support the conclusion that $CO_2$ added to the inspired air inhibits, and in higher concentration blocks the neurovascular response, independently of changes in brain tissue pH and perfusion. That disruption of $CO_2/HCO_3^-$ transport in astrocytes completely prevents the neurovascular coupling response stands in a stark contrast to the data obtained in the majority of the preceding studies, where only partial inhibition of the response was commonly observed[5].

The data obtained in our experiments are consistent with the results of a recent study conducted in humans which showed significant reductions in the magnitude and kinetics of the neurovascular response during experimental alterations in arterial $CO_2$[24]. However, earlier human studies showed that while systemic hypercapnia increases the baseline CBF, it has no or little effect on the absolute magnitude of the neurovascular coupling response. In one study conducted in human volunteers[25], 5% $CO_2$ was added to the inspired air to increase $P_{ET}CO_2$ by ~6 mmHg. This treatment increased basal CBF by ~50% and had no effect on the expression of the neurovascular response to visual stimuli[25]. Similar experimental treatment (5% inspired $CO_2$) was applied in another human study resulting in $P_{ET}CO_2$ increase by ~10 mmHg[26]. This level of hypercapnia was associated with a significant (by ~30%) reduction of the visually-evoked BOLD response (no CBF data were reported)[26]. Assuming that within a physiological range there is a linear relationship between brain tissue $PCO_2$ and perfusion, we may expect that in conditions of mild/moderate hypercapnia activity-dependent increases in CBF (driven by $CO_2$ produced as a result of increased neuronal activity) can still develop on top of the elevated baseline CBF. If this hypothesis is correct, then much higher concentration of exogenous $CO_2$ (compared to what can be applied in human studies) would be required to fully saturate the $CO_2$-sensitive vasodilatory mechanism and occlude the actions of

$CO_2$ produced by brain neurons. The experiments of this type were performed in the present study using anaesthetized animals. In agreement with the data obtained in humans[24,26], the results of our study show partial inhibition of the neurovascular response when 5% $CO_2$ was given in the inspired air, whilst higher levels of hypercapnia (10% inspired $CO_2$) were required to prevent the development of the neurovascular response.

As most of glucose metabolized by the brain tissue ends up as $CO_2$ in neurons, generation of $CO_2$ is proportional to neuronal activity and energy use. Being readily diffusible, $CO_2$ is an ideal molecule to rapidly signal changes in the activity of brain circuits. We originally hypothesized that $CO_2$ mediates the component of the neurovascular response (30–40%) which is insensitive to pharmacological or genetic blockade of the feed-forward mechanisms, suggested by the preceding studies[5]. However, our systematic review and analysis of published data obtained in the experimental animal and human studies of the mechanisms underlying cerebrovascular $CO_2$ sensitivity showed that the same key signaling pathways that mediate the neurovascular coupling response[5], also mediate the effects of $CO_2$ on brain blood flow and involve NO and cyclooxygenase products (Fig. 1). In particular, blockade of neuronal NOS reduces the neurovascular response by 64%[5] and inhibits the $CO_2$-induced cerebrovascular dilations by 40% (Fig. 1). Blockade of cyclooxygenase with indomethacin (note potential off-target effects of the drug[27]) inhibits the neurovascular coupling response and cerebrovascular $CO_2$ reactivity by exactly the same degree—44% (ref. 5 and Fig. 1). Partial inhibition of the neurovascular response with indomethacin was also observed in this study, as discussed above (it is important to note that despite hundreds of relevant studies, the cellular and molecular mechanisms of $CO_2$ sensing, leading to the recruitment of the identified signaling mechanisms, remain unclear).

It follows that the hypothesis of $CO_2$ mediating the neurovascular coupling response is fully compatible with the current understanding of the mechanisms underlying the neuronal activity-dependent control of brain arterioles and capillaries by arterial smooth muscle cells and pericytes[7]. Dilation of arterioles in response to increases in neuronal activity have been suggested to be mediated by NO[7], while capillary responses have been shown to be triggered by a signaling pathway that involves the release[28] and actions of ATP, leading to $Ca^{2+}$−dependent activation of cyclooxygenase-1 and release of prostaglandins by astrocytes[7]. Although the mechanisms and sources of neuronal activity-induced ATP release were not described in these earlier studies, ATP is well known to be released by astrocytes and

possibly other brain cells in a $CO_2$-sensitive manner[20,29–31]. There is evidence that $CO_2$ can be sensed directly (i.e. not via a proxy of associated pH changes) by hemichannels which belong to the beta-connexin family (such as connexin 26, connexin 30, and connexin 32) and increase their open probability (allowing egress of ATP) proportionally to the concentration of $CO_2$ which forms carbamate bridges between subunits forming the channel[31–33]. ATP triggers $Ca^{2+}$ responses in astrocytes via metabotropic and ionotropic receptors[7,19], but can also cause depolarization and increase firing of hippocampal and cortical interneurons via $P2Y_1$ receptor activation[34]. This would be expected to result in enhanced generation of NO by nNOS-expressing cells that project to brain parenchymal arterioles. We hypothesize, therefore, that signaling in the neurovascular interface, mediated by NO and cyclooxygenase products, is ultimately initiated and driven by the actions of metabolically produced $CO_2$.

Accordingly, based on the experimental data obtained in this study we suggest a reconsideration of the currently prevailing view that the neurovascular response is mediated exclusively by neurotransmitters and/or $K^+$ released by active neurons[3,6,7]. We propose a unifying model of the neurovascular coupling response which combines neuronal activity-dependent 'feed-forward' and $CO_2$-mediated metabolic feed-back mechanisms working in concert to maintain optimal nutrient and oxygen supply to support the neuronal metabolic needs, but most importantly for the purpose of effective removal of large quantities of $CO_2$ generated by the brain.

## Methods

All animal experiments were performed in accordance with the European Commission Directive 2010/63/EU (European Convention for the Protection of Vertebrate Animals used for Experimental and Other Scientific Purposes) and the UK Home Office (Scientific Procedures) Act (1986) with project approval from the Institutional Animal Care and Use Committees of the University College London and Centro de Estudios Científicos. The animals were group-housed and maintained on a 12-h light cycle (lights on 07:00) and had ad libitum access to water and food.

**General animal preparation.** Young adult male Sprague-Dawley rats (280–350 g; 2–3 months old; Charles River, UK) were used in the imaging experiments. The animals were anesthetized with α-chloralose (induction: 100 mg kg⁻¹, maintenance: 30 mg kg⁻¹ h⁻¹, i.v.). The femoral artery and vein were cannulated for continuous monitoring of the arterial blood pressure and the administration of anesthetic, respectively. Adequate depth of anesthesia was confirmed by the stability of arterial blood pressure and heart rate recordings which did not show responses to a paw pinch. The animal was intubated and mechanically ventilated with oxygen-supplemented air using a small rodent ventilator (tidal volume ∼0.8 ml per 100 g of body weight; ∼60 strokes min⁻¹). Neuromuscular blockade was established following administration of gallamine (induction: 5 mg kg⁻¹, maintenance 2 mg kg⁻¹ h⁻¹, i.v.). Arterial $PO_2$, $PCO_2$, and pH were measured regularly and kept within the physiological ranges ($PO_2$ 100–120 mmHg; $PCO_2$ 35–40 mmHg; and pH 7.35–7.45) by adjusting the tidal volume and/or ventilator frequency as well as the amount of supplemental oxygen. Body temperature was maintained at 37.0 ± 0.5 °C using a servo-controlled heating pad.

**Functional magnetic resonance imaging.** fMRI was used to record the neurovascular coupling response in the somatosensory cortex. We implemented an ASL sequence with T2*-weighted imaging for combined measurement of local CBF (in ml 100 g⁻¹ min⁻¹) and BOLD signal changes in the S1FL region of the cortex. The measurements of absolute changes in CBF were essential to address the objectives of this study as the BOLD signal reflects an unknown combination of changes in local CBF, blood volume, and cerebral metabolic rate of oxygen. ASL measurements provided absolute values of cerebral perfusion in order to control for possible confounding effects of the experimental manipulations on the expression of the neurovascular response.

fMRI was performed using a 9.4 T Agilent horizontal bore scanner (Agilent)[28,35]. The animal was anaesthetized and instrumented as described above (see the section "General animal preparation") and then transferred to the MRI scanner bed. The head was secured with ear and incisor bars. A 72 mm inner diameter volume coil was used for radio frequency transmission and signal was received using a 4-channel array head coil (Rapid Biomedical). First, a high-resolution anatomical reference scan was acquired using a fast spin echo sequence (TR/TE_eff = 3100/48 ms, ETL = 8, matrix size = 256 × 256, FOV = 35 mm × 35 mm, 30 slices, 1 mm slice thickness). The anatomical reference image was used to manually position the single functional coronal imaging slice (see below) to be centered on the S1FL region.

Resting CBF and CBF/BOLD responses to somatosensory stimulation were recorded using a flow-sensitive alternating inversion recovery (FAIR) ASL sequence, with concurrent BOLD T2*-weighted imaging, using the following sequence parameters: single shot gradient echo EPI readout, TR = 5000 ms, inflow time (TI) = 2000 ms, matrix size = 64 × 64 voxels, FOV = 35 × 35 mm, TE = 10 ms, single slice (thickness = 2 mm), inversion pulse bandwidth = 20,000 Hz. CBF maps were generated by fitting the data to previously established models[36], where T1 in the cortex was assumed to be 1.7 s, the arterial transit time to be 0.3 s and the temporal duration of the tagged bolus to be 2 s[37], based on previous measurements using the identical experimental conditions and the equipment.

**fMRI data analysis.** CBF and BOLD time-series data were extracted from manually drawn regions of interest (ROIs) based on the baseline functional data and were fixed for all the subsequent conditions. The BOLD signal was taken from the alternate 'control' images in the FAIR ASL acquisition. In order to partially correct the ASL (ΔM) signal for marked BOLD T2*-weighted signal changes that occur when the forepaw stimulus begins/ends at a time between the labeled and control acquisitions[38], the labeled image directly after the stimulus onset was substituted by the previous labeled image (captured during the baseline period). Similarly, the control image taken directly after the cessation of the stimulus was substituted by the previous control image (captured during the stimulation period). For the CBF and BOLD time-course data, occasional large amplitude artefacts in the recorded signal due to hardware instability were removed using an automated de-spiking algorithm. BOLD signal responses are expressed as changes from the average of pre-stimulus baseline. CBF responses were assessed by calculating the area under the curve (AUC) for each experimental condition. In order to generate maps of BOLD signal responses for visualization purposes, the control images were spatially smoothed (0.5 mm FWHM Gaussian kernel) and first level analysis of each time-series using an on/off regressor derived from the applied forepaw stimulus paradigm (and convolved with the standard HRF (SPM)) was applied to generate the statistical maps. In order to visualize the BOLD activation maps to forepaw stimulation, a minimum threshold of $p < 0.0001$ with a cluster size of >5 voxels was applied.

**Fast scan cyclic voltammetry.** Tissue $PO_2$ and evoked neuronal activity in the S1FL region of the cortex were recorded using fast cyclic voltammetry[22]. The animal was anaesthetized and instrumented as described above (see the section "General animal preparation"). The head was secured in a stereotaxic frame with ear and incisor bars. After a small craniotomy (∼1 mm²) to allow access to the somatosensory cortex, carbon fiber microelectrode (CFM; diameter 7 μm) was advanced into the S1FL cortex until evoked field potentials were recorded in response to the electrical stimulation (1 Hz) of the contralateral forepaw. A series of voltage ramps (200 V s⁻¹) from 0 to −1 V were applied to the CFM at a frequency of 2 Hz. The resulting current was amplified, digitized and recorded for offline isolation of faradaic current to determine changes in tissue $PO_2$. The CFM recordings were continuously switched between current amplification and voltage amplification to allow near-simultaneous detection of the evoked extracellular potentials (voltage). CFM recordings were acquired using Power1401 interface and analyzed offline using Spike2 software (version 7; Cambridge Electronic Design).

**Recordings of extracellular pH and neuronal activity.** Changes in extracellular pH and evoked neuronal activity in the S1FL region of the cerebral cortex were recorded using fast scan cyclic voltammetry in a separate cohort of male Sprague-Dawley rats (2–3 months old, $n = 27$) under the experimental conditions that were identical to that during the imaging experiments (see the section "General animal preparation"). Brain tissue pH changes were recorded continuously during each of the experimental challenges (provision of $CO_2$ in the inspired air and/or administration of pharmacological agents) and expressed as averages of values recorded during a 1 min period. Somatosensory stimulation-induced neuronal responses were assessed by integration of the evoked volley of extracellular potentials with the baseline noise subtracted. Recordings obtained during three trains of forepaw stimulation were pooled for the analysis.

**Conditional NBCe1 knockdown in astrocytes.** To induce conditional NBCe1 knockdown in astrocytes, mice carrying a loxP-flanked NBCe1 allele (NBCe1flox/flox)[39] were crossed with the mice expressing an inducible form of Cre (CreERT2) under the astrocyte-specific GLAST promoter[40]. All animals were on C57Bl/6J genetic background. Tamoxifen (100 mg kg⁻¹) dissolved in corn oil was given to NBCe1flox/flox:GLASTCreERT2/+ male mice ($n = 27$) at postnatal week 7 and the expression level of NBCe1 was examined 6 weeks after the tamoxifen treatment. Breeding was organized through PCR genotyping obtained from tail DNA biopsies. Astroglial recombination specificity of GLAST-CreERT2 mice has been reported in several prior studies[40,41]. Development and characterization of this animal model had been described in detail previously[19]. Littermate NBCe1flox/flox:GLAST-CreERT2/+ mice injected with the vehicle (oil, $n = 18$) and NBCe1flox/flox:GLAST+/+ mice injected with tamoxifen ($n = 17$) were used as controls. At the end of the experiments, 4 animals from each of the experimental groups were given an anesthetic overdose, the brains were removed, fixed in 4% paraformaldehyde, sliced (7 μm), and NBCe1 expression was detected by immunostaining using primary mouse monoclonal anti-NBCe1 antibodies (1:50; Santa Cruz Biotechnology, Cat #

sc-515543, Lot # I2216) and secondary anti-mouse antibodies Dylight 488 (1:500; ThermoFisher, Cat #35502). The animals were between 3 and 4 months old at the time of the experiments.

In the open field test, the mouse was placed in the center of an arena (40 × 40 × 30 cm) equipped with infrared sensors (Med Associated Inc.). The behavior of the animal in the open field was recorded for 10 min. Respiratory activity was assessed using whole-body plethysmography[42,43]. The animal was placed in a recording chamber (~200 ml) which was flushed continuously with a humidified mixture of 79% nitrogen and 21% oxygen at a rate of ~500 ml min$^{-1}$ (temperature 22–24 °C). The animal was allowed ~30 min to acclimatize to the chamber environment before measurements of baseline ventilation were taken. Hypercapnia was induced by adding $CO_2$ to the chamber gas mixture up to a level of 5% and 10% (lowering $N_2$ accordingly) for 5 min at each $CO_2$ level. The measurements of the ventilatory variables (respiratory rate and tidal volume) were obtained during the last 2 min before exposure to the $CO_2$ stimulus and during the 2 min period near the termination of each stimulus, when breathing had stabilized.

## Experimental protocols

*Experiment 1: The effect of surplus of exogenous $CO_2$ on the neuronal activity-dependent increases in local cerebral blood flow in the somatosensory cortex.* In rats ($n = 7$), anaesthetized and instrumented as described above (see the section "General animal preparation"), activation of somatosensory pathways leading to robust CBF and BOLD signal responses in the S1FL cortical region was achieved by electrical stimulation of the forepaw (300 μs pulse width, 3 Hz, 1.5 mA), applied using bipolar subcutaneous electrodes delivering fixed-current pulses from an isolated stimulator (Digitimer DS3). Three forepaw stimulations 60 s in duration each, repeated with 60 s inter-stimulus intervals and a 60 s baseline period were applied with CBF and BOLD signal responses to stimulations averaged and compared between each of the following experimental conditions: (1) control condition: animals were ventilated with oxygen-enriched room air and ventilation parameters adjusted to ensure the arterial $PO_2$, $PCO_2$ and pH were maintained within the physiological ranges, as described above; (2) 5% inspired $CO_2$ condition: 5% $CO_2$ was added to the inspired gas mixture; (3) 10% inspired $CO_2$ condition: 10% $CO_2$ was added to the inspired gas mixture; (4) brain acidosis condition: $CO_2$ was withdrawn from the inspired air and acetazolamide was given systemically (10 mg kg$^{-1}$, i.v.) to inhibit carbonic anhydrase (CA) causing a decrease in brain pH; (5) 5% inspired $CO_2$ in conditions of CA inhibition: in the continuing presence of systemic acetazolamide, 5% $CO_2$ was added to the inspired gas mixture; (6) 10% inspired $CO_2$ in conditions of CA inhibition: in the continuing presence of systemic acetazolamide, 10% $CO_2$ was added to the inspired gas mixture. Each condition was established for at least 10 min prior to the electrical stimulation of the forepaw to trigger the neurovascular response in the S1FL region. The duration of the experiment was <90 min.

*Experiment 2: The effect of surplus of exogenous $CO_2$ on the neuronal activity-dependent increases in local cerebral blood flow in the somatosensory cortex before and after the baseline (normocapnic) level of CBF is restored pharmacologically.* To counteract the vasodilatory effect of $CO_2$ and lower the CBF to the baseline level recorded at normocapnia, the rats ($n = 19$) were given 10% $CO_2$ in the inspired air and then treated systemically with two pharmacological agents known to reduce the CBF—caffeine and indomethacin. These drugs were applied either alone or in combination, in three separate experiments involving a sequence of experimental conditions described below. Three forepaw stimulations 60 s in duration each, repeated with 60 s inter-stimulus intervals and a 60 s baseline period were applied with CBF and BOLD signal responses to stimulations averaged and compared between each of the following experimental conditions studied sequentially: (1) *control condition*: animals were ventilated with oxygen-enriched room air and ventilation parameters adjusted to ensure the arterial $PO_2$, $PCO_2$ and pH were maintained within the physiological ranges, as described above; (2) 10% inspired $CO_2$ condition: 10% $CO_2$ was added to the inspired gas mixture; (3) 10% inspired $CO_2$ condition with baseline (normocapnic) CBF restored pharmacologically: either caffeine, indomethacin, or a combination of caffeine and indomethacin (both drugs at 10 mg kg$^{-1}$, i.v.) were given concurrently with 10% $CO_2$ in the inspired gas mixture; (4) systemic caffeine/indomethacin condition: $CO_2$ was withdrawn from the inspired air in conditions of continuing systemic caffeine/indomethacin action. The duration of the experiment was <90 min.

A separate group of animals ($n = 9$) was studied under the same sequence of experimental conditions but when 10% $CO_2$ was given in the inspired gas mixture (conditions 2 and 3), the electrical current applied to the forepaw was increased to 3 mA in order to compensate for the $CO_2$-induced inhibition of the neuronal activity, as described in the "Results" section.

*Experiment 3: The effect of NBCe1 knockdown in astrocytes on neuronal activity-dependent increases in local cerebral blood flow in the somatosensory cortex.* Fast scan cyclic voltammetry was used to record the neurovascular response in the S1FL region of the somatosensory cortex in mice. The expression of the neurovascular response was assessed from the recordings of changes in brain tissue $PO_2$, previously reported to accurately match BOLD responses to somatosensory stimuli[22]. Mice (3–4 months old) were anesthetized with a mixture of ketamine/xylazine (100 mg kg$^{-1}$/10 mg kg$^{-1}$, i.p.). The head was secured in a stereotaxic frame. Oxygen-enriched room air was supplied to the animal breathing unaided throughout the experiment.

Electrical forepaw stimulations (300 μs pulse width, 3 Hz, 1.5 mA) were applied to activate somatosensory pathways and evoke the neurovascular response in the S1FL region. S1FL tissue $PO_2$ changes evoked by three stimulations (20 s long) were averaged within each animal to obtain a profile of the response for each subject. Peak of the response (difference in $PO_2$ between the value taken at the end of the stimulation period and the pre-stimulus baseline, expressed in mmHg) and integral (area under the curve, AUC) were compared between the astrocyte specific NBCe1 deficient (NBCe1$^{flox/flox}$:GLAST$^{CreERT2/+}$ animals treated with tamoxifen, $n = 7$) and two control groups of mice (NBCe1$^{flox/flox}$:GLAST$^{CreERT2/+}$ animals given vehicle, $n = 7$; and NBCe1$^{flox/flox}$:GLAST$^{+/+}$ mice treated with tamoxifen, $n = 8$).

**Drugs**. Acetazolamide and indomethacin (Cambridge Bioscience) were dissolved in 100% DMSO (100 mg ml$^{-1}$) and administered i.v. (10 mg kg$^{-1}$). Caffeine (Sigma) was dissolved in saline and administered i.v. (10 mg kg$^{-1}$). The maximum volume of administered DMSO was 100 μl kg$^{-1}$, previously shown to have no effect on the cerebrovascular responses[44].

**Data analysis**. Statistical analysis of the data was performed using GraphPad-Prism software (version 8). The data are reported as means ± SEM or as box-and-whisker plots. Normal distribution of data was determined by the Shapiro–Wilk test. Grouped data were analyzed using one-way or mixed-model ANOVA or Kruskal–Wallis test (for non-normally distributed data) when comparing data between more than two groups. Post-hoc analysis was performed corrected for multiple comparisons using Tukey's, Dunn's or Holm–Sidak method. Two-tailed Wilcoxon's matched-pairs signed rank test or Mann–Whitney-$U$ test were used to compare the paired and un-paired data between two experimental groups/treatments. Details of the statistical tests applied are provided within the figure legends.

**Systematic review of the literature data on the mechanisms of cerebrovascular response to $CO_2$**. The existence of multiple (sequential, parallel, overlapping, and interacting) signaling pathways that can potentially mediate the effects of $CO_2$ on cerebral vasculature is suggested by numerous preceding studies. The relative significance of these pathways in mediating cerebrovascular responses to $CO_2$ was evaluated by systematic review and analysis of published data obtained in the experimental animal and human studies that targeted all hypothesized mechanisms either pharmacologically or genetically. Initial searches returned hundreds of relevant studies. Studies meeting the criteria outlined next were selected for meta-analysis. Our primary outcome measure was the percent reduction of the cerebrovascular response to $CO_2$ recorded in in vivo animal models and in studies involving human participants. A minimum of three experimental studies targeting the same signaling pathway were required for inclusion. Selection criteria were met by 131 primary sources, reporting the data obtained under 214 different experimental conditions (Supplementary Table 1 and Supplementary References). Experimental conditions were grouped into categories that included studies targeting mechanisms mediated by NO, cyclooxygenase products, adenosine, and those aimed at blocking the other or several targets simultaneously. Means were taken from the control and experimental group data and the effect of a pharmacological agent or genetic manipulation was expressed as percentage change from the control response. The individual data points were then pooled per targeted signaling pathway and plotted with their respective 95% confidence intervals (Fig. 1). This analysis standardizes the relative effectiveness of blocking each target or a combination of targets, allowing direct comparisons.

**Reporting summary**. Further information on research design is available in the Nature Research Reporting Summary linked to this article.

## Data availability
The data that support the findings of this study are available from the corresponding authors upon request. Source data are provided with this paper.

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

## Acknowledgements

This work was supported by The Wellcome Trust (A.V.G.; refs: 200893 and 223057), Fondecyt Iniciación Grant 11190678 (I.R.) and ANID-BMBF 180045 grant (L.F.B.). A.V.G. was supported by Wellcome Senior Research Fellowship (ref: 200893). J.A.W. was supported by Wellcome Trust/Royal Society Sir Henry Dale Fellowship (ref: 204624). CECs is funded by the Chilean Government through the Centers of Excellence Base Financing Program. We thank Professor Gary E. Shull (Cincinnati, USA) for providing NBCe1$^{flox/flox}$ mice and Professor Frank Kirchhoff (Hamburg, Germany) for providing GLAST-CRE ERT2 mice. We thank Dr Bredford Kerr (Santiago, Chile) for his help in running the behavioural studies. We are also grateful to Professor David Attwell for his comments on the first version of the manuscript.

## Author contributions

A.V.G. conceived and directed the project; A.V.G., P.S.H., J.A.W., and I.R. designed research; P.S.H., J.A.W., S.N., I.N.C., S.M.T., P.A.C., A.H., I.R., and A.V.G. performed research; L.F.B. and M.F.L. contributed unpublished reagents/analytic tools; P.S.H., J.A.W., S.N., I.N.C., S.M.T., A.H., I.R., and A.V.G. analyzed data; A.V.G. and P.S.H. wrote the paper. All authors revised the article critically for important intellectual content.

## Competing interests

The authors declare no competing interests.
