## [Peer Review File · Nature Communications]

CO2 signaling mediates neurovascular coupling in the cerebral cortexREVIEWER COMMENTS

Reviewer #1 (Remarks to the Author):

The manuscript by Hosford titled "CO₂ signalling mediates neurovascular coupling in the cerebral cortex" investigates whether local generation of CO₂ (or pH changes) in the brain by neurons rapidly cause vasodilation of nearby microvasculature which is responsible for a sensory-evoked BOLD fMRI signal. This work is important because CO₂ is fundamental to cerebral blood flow regulation yet through mechanisms that are somewhat unclear and/or poorly contextualized within the physiology of the system. CO₂ was historically not thought to be involved in functional hyperemia, instead perhaps playing a role constantly for basal blood flow regulation, or via systemic increases in blood CO₂, which is well appreciated to dilate large cerebral blood vessels. The main idea is saturate the 'local CO₂ control system' by saturating the system with systemic CO₂. While ceiling effects are the biggest concern here, the authors attempt to control for this with systemically administered vasoconstrictors. The baseline shifts in CBF are shown upfront in the data, which is strength of the study. Using different techniques (direct O₂ sensors) the authors then explore a potential mechanisms of CO₂ hyperemia via the sodium bicarbonate co-transporter in astrocytes using cre-lox knockout. This work is also important because exactly how astrocyte contribute to cerebral blood flow regulation is unclear, especially regarding Ca²⁺ dependent mechanisms. This work potentially describes a new, maybe Ca²⁺ independent, form of CBF control via astrocytes. All the data are clearly and beautifully presented. While overall an exciting concept and study, I have a number of significant concerns with the some of the general approaches to target this question, how it fits into existing literature on this topic, as well as the data interpretation.

Major

1) Controlling for baseline blood flow with systemic indo and caffeine is not very selective. Indo blocks COX but also blocks PKA (PMID: 214715), and can directly affect functional hyperemia too (more below), Caffeine has numerous targets on the adenosine system and on Ca²⁺ stores in many cell types. Indo blocks CO₂ reactivity (PMID: 643149) and both drugs would have systemic effects on blood pressure too. More specific concerns are below, but here, it would have been much more elegant if the authors selectively controlled for tone in the forepaw region using vascular DREADDs previously delivered using locally injected AAV, followed by systemic administration of CNO during the imaging experiment. Perhaps via mural cell expression of Gq or Gi DREADD to decrease blood flow to the level they need. While this would have taken the generation of a custom virus and some validation, I don't think this technique should be out of reach, and may be necessary to achieve clear conclusions about the results.

2) Indo and caffeine effects are hard to interpret. Looking at the data, even though they restore baseline blood flow, they appear to block functional hyperemia, at least in the high CO₂ condition. More specifically, in 10% CO₂, using stronger stimulation (3mA) there is a BOLD response, but in the presence of indo and caffeine, still in 10% CO₂, baseline has decreased but 3mA stim produces no BOLD response. This suggests indo and caffeine are not just controlling for baseline blood flow, but also indirectly block functional hyperemia in this high CO₂ state. It is appreciated that indomethacin reduces blood flow increases to systemic hypercapnia, which is likely why baseline blood flow comes down, yet there could be additional effects of indo and caffeine on functional hyperemia itself only in this high CO₂ state, unrelated to the presumed CO₂-mediated neurovascular coupling. How do the authors explain this loss of the 3mA stim induced BOLD signal in indo and caffeine?

3) Or perhaps the indo + caffeine block I highlighted above relates to a direct effect on CO₂ NVC and there is previous literature here that the authors do not adequately address. Indo should block COX1 and a previous report attempted to connect locally produced CO₂ induced vasodilation via COX1 activation in astrocytes. That COX1 is involved in cerebral vasodilation to systemic hypercapnic was clearly shown in a classic general knockout study (PMID: 11282894), though COX-1 here is not involved in sensory induced functional hyperemia. More recently, the astrocyte COX1 pathway was shown to be recruited in response to locally applied CO₂, which elevated astrocyte Ca²⁺ to cause a

PG dependent dilation (PMID: 28137973). These previous studies should be at least be mentioned or discussed if space permits.

4) It is unclear if acetazolamide would restore functional hyperemia in hypercapnia or not because baseline is very high in acetazolamide alone, which is raised even further in 5% and 10% CO₂. Perhaps a ceiling effect occurs in this condition? The authors did not constrict the vessels in acetazolamide to restore baseline and then test forepaw stimulation.

5) It is a little unexpected that indomethacin doesn't block functional hyperemia in normocapnia. There are several papers that show this (PMID: 17655958; PMID: 24262992)

6) The use of ASL for baseline and stim induced BOLD fMRI is, on one hand commendable by demonstrating a genuine change in O₂ in the forepaw region, but on the other hand, tricky to place in context of previous work. Though I am not an expert here, MR work typically necessitates long stimulation paradigms to capture the data (here 1.5 min of sensory stimulation). Unfortunately, the temporal resolution usually employed in optical imaging studies of functional hyperemia is much faster. Functional hyperemia, driven by neural mediators, can be up and down in several seconds, thus it would be more interesting to see the effects of CO₂ on this time scale. Maybe locally produced CO₂ is important on the time scale of several tens of seconds, but what about in the first few seconds? Can the authors comment on this?

7) I really like the idea to knockout NBCe1 in astrocytes as a mechanistic link between CO₂ and vasodilation. This is a novel and exciting experiment. However, the authors did not setup this experiment properly: there is no control for tamoxifen. They need a third group that receives tamoxifen in the same animals that are negative for cre or the floxed gene. Otherwise it is impossible to know whether the effect they report is caused by tamoxifen itself, rather than gene knockdown.

8) What is baseline CBF like in the eNBCe1 knockout? This is equally important as the other data. If baseline is up significantly, then baseline needs to be controlled for to ensure there is no ceiling effect.

9) Why is the NBCe1 knockout work not done using ASL MR like the rest of the study? Instead the authors use local oxygen partial pressure measurements. It would be nice to see correspondence in the data between the different experiments using a single measurement.

10) The immuno images for NBCe1 do not look comparable to each other, and they also do not look convincing that the authors have detected the protein of interest on astrocytes and they have successfully knocked it down. A more convincing immuno-fluorescence or biochemical approach must be shown for this knockdown experiment.

Reviewer #2 (Remarks to the Author):

To the authors:

Hosford et al. investigate the role of CO₂ in neurovascular coupling by saturating brain CO₂-sensing mechanisms with inspired exogenous CO₂, reasoning that this will nullify responses of blood vessels to further increases in CO₂ that result from neuronal activation and increased metabolic activity. Using this paradigm, they find that a CO₂ is a major contributor to cerebral blood flow responses to neuronal activity, independent of tissue pH. The paper is well written, the data are quite compelling and the main effect of CO₂ on CBF/BOLD responses is striking, and argue for the integration of this mechanism into our current understanding of neurovascular coupling mechanisms. I have the following questions/concerns at this point:

i. In order to compensate for the possibility that changes in basal CBF are confounding the interpretation of the data by lowering the vasodilatory reserve of vessels, the authors utilize caffeine and indomethacin to return CO₂-induced elevations in CBF back to baseline levels. The data indicate that under these conditions, CO₂ still eliminates the CBF response to forepaw stimulation. Following this, a return to normocapnia leads to normal CBF/BOLD responses, suggesting that the pathways blocked by indomethacin and caffeine do not participate in neurovascular coupling. However, this finding contradicts a number of studies that have found a substantial contribution of the cyclooxygenase pathway, targeted by indomethacin, to neurovascular coupling in both rodents and humans in a range of preparations (see references below). Why is no contribution of these pathways observed under the present conditions? The authors do not note the timing of this experiment in the methods, i.e. the duration of the rest periods between the different intensities of electrical forepaw stimulation, and the length of time that was allowed for the washout of inhaled CO₂ upon the return to normocapnia. How long were these periods? Could a long wait have led to the metabolism of indomethacin to ineffective concentrations? What is the effect of these compounds on the CBF/BOLD response under normocapnic conditions immediately after administration of indomethacin/caffeine, when they are at their maximal plasma concentrations? Since these compounds may block pathways that are involved in neurovascular coupling, I suggest that a more 'neutral' drug choice for baseline correction may make data interpretation more straight forward. With this in mind, what is the effect of hypercapnia on CBF/BOLD responses when blood flow is instead corrected using phenylephrine (see Ogoh et al. 2011 below)? The corollary of this is whether the authors can still reliably increase blood flow well above the already elevated level under 10% CO₂? This could be addressed by administering an L-type voltage-gated calcium channel blocker in the presence of CO₂, for example, which should still produce a robust increase in blood flow.

Takano T, Tian GF, Peng W, Lou N, Libionka W, Han X, Nedergaard M. Astrocyte-mediated control of cerebral blood flow. *Nat Neurosci.* 2006 Feb;9(2):260-7. doi: 10.1038/nn1623. Epub 2005 Dec 25. PMID: 16388306.

Lecrux C, Toussay X, Kocharyan A, et al. Pyramidal neurons are "neurogenic hubs" in the neurovascular coupling response to whisker stimulation. *J Neurosci.* 2011;31(27):9836-9847. doi:10.1523/JNEUROSCI.4943-10.2011

Lacroix A, Toussay X, Anenberg E, Lecrux C, Ferreirós N, Karagiannis A, Plaisier F, Chausson P, Jarlier F, Burgess SA, Hillman EM, Tegeder I, Murphy TH, Hamel E, Cauli B. COX-2-Derived Prostaglandin E₂ Produced by Pyramidal Neurons Contributes to Neurovascular Coupling in the Rodent Cerebral Cortex. *J Neurosci.* 2015 Aug 26;35(34):11791-810. doi: 10.1523/JNEUROSCI.0651-15.2015. PMID: 26311764; PMCID: PMC6705452.

Ogoh S, Sato K, Fisher JP, Seifert T, Overgaard M, Secher NH. The effect of phenylephrine on arterial and venous cerebral blood flow in healthy subjects. *Clin Physiol Funct Imaging.* 2011 Nov;31(6):445-51. doi: 10.1111/j.1475-097X.2011.01040.x. Epub 2011 Jul 12. PMID: 21981455.

ii. A number of studies in the past have used 5% CO₂/95% O₂, or 5% CO₂/20% O₂/balance N₂ to gas solutions which are used to perfuse brain slices and other preparations to study neurovascular coupling. It seems that, according to the present data, these conditions should greatly blunt neurovascular coupling/vasodilations, yet these are still readily observable under these conditions. How should the data from the present study be integrated with these findings?

iii. Neurovascular coupling is characterized as resulting from communication between neurons and astrocytes with vascular smooth muscle cells and pericytes, and the contribution of the endothelium appears to have been overlooked. Our group also recently provided direct evidence that parenchymal cells communicate with the endothelial cells that compose the capillaries.

Longden TA, Dabertrand F, Koide M, et al. Capillary K⁺-sensing initiates retrograde hyperpolarization to increase local cerebral blood flow. *Nat Neurosci.* 2017;20(5):717-726. doi:10.1038/nn.4533

iv. The data indicate that CO₂ makes a major contribution to neurovascular coupling and that with CO₂-sensing mechanisms saturated, neurovascular coupling does not proceed. Emerging evidence from a number of labs supports important contributions of a range of mediators to this process,

including nitric oxide, K⁺, prostaglandins, and epoxyeicosatrienoic acids, among others, as the authors acknowledge. According to the data herein, it appears that CO₂ sensing thus underpins all (or a very large fraction) of these diverse mechanisms. It is important to discuss how, at a mechanistic level, CO₂ sensing might underlie these, as this is not clarified in detail in the present version of the manuscript. The authors do note that ATP release may be a CO₂-sensitive process, and this discussion point could be expanded to provide a more global view of CO₂'s contribution to other neurovascular coupling mechanisms as well.

Best regards,

Tom Longden

Reviewer #3 (Remarks to the Author):

The past 10 years have provided numerous studies showing quite compelling evidence that neuronal activity causes regulation of blood flow via the release of vasoactive molecules from either astrocytes or pericytes. These signals include prostaglandins, 20-HETE, ATP and K⁺ in a process called functional hyperemia. This phenomenon is underlying the BOLD effect utilized in fMRI studies to interrogate brain function.

The current paper challenging this dogma, namely that functional hyperemia involves glia, but rather the authors propose that it is entirely due to local CO₂ changes as a result of oxidative metabolism. Unlike most prior studies, the authors utilize MRI and the BOLD signal as a functional readout, equating an increase in regional bold signal with vasodilation and a decrease with vasoconstriction. The region of interest is the somatosensory cortex and the stimulus used is electrical forepaw stimulation. The major experimental argument that leads the authors then to propose that CO₂ is the principle vasoregulator is the fact that inhalation of super-physiological concentrations of CO₂, namely 10% leads to a complete loss to changes in blood flow in response to electrical forepaw stimulation. However this level of CO₂ causes maximal vessel dilation and increases cerebral blood flow almost 3 fold.

One argument one could make therefore, right of the bat, is that no further vasodilation would be possible for vessels already maximally dilated.

To counterargue this, the authors then give the animals the vasoconstrictors indomethacin and caffeine in the presence of 10% CO₂ and again see no BOLD response to forepaw stimulation. It is known that release of prostaglandin can dilate pre-constricted blood vessels in brain slices. However, this experiment does not proof that CO₂ release from neurons, bypassing glia cause vasodilation.

Unfortunately the key experiment to substantiating the author' claim that CO₂ released from neurons is necessary and sufficient to regulate blood flow is missing. It is often called the "block" experiment and would have to block CO₂ release from neurons in spite of forepaw stimulation. In other words disrupting the signaling molecule should no longer lead to changes in blood flow.

Also, the approach used in this study does not allow to discern whether change in blood flow occur at the capillary level, in penetrating arterioles and feeding arteries. The studies that so nicely implicate astrocytes as the purveyor of the signal were done with cellular resolution showing activation of single astrocytes causing time-synchronized vasodilations. It maybe necessary to repeat these studies in the presence of 10% CO₂ and show that now, astrocytes no longer participate. Might it even be possible that it is the astrocytes that sense the CO₂ and ultimately the response, while ascribed to CO₂ still requires glia as an intermediate?

Changing dogma carries a significant burden of proof. While I find these studies compelling, I am not convinced that they are strong enough yet to substantiate the claim that countless other studies got it wrong.

Manuscript ID: NCOMMS-20-46461-T
Responses to the referees' comments

We would like to thank all three reviewers and the Editors of *Nature Communications* for their time taken to evaluate our submission and overall positive assessment of our work. We are grateful for the detailed and constructive comments provided and delighted to have an opportunity to re-submit our work. In this revised submission we include a comprehensive analysis of the literature data describing the mechanisms underlying the effects of CO₂ on cerebral vasculature, and additional experimental data requested by the reviewers. We provide a full response to all the criticisms raised and submit a thoroughly revised manuscript.

Reviewer #1:

The manuscript by Hosford titled "CO₂ signalling mediates neurovascular coupling in the cerebral cortex" investigates whether local generation of CO₂ (or pH changes) in the brain by neurons rapidly cause vasodilation of nearby microvasculature which is responsible for a sensory-evoked BOLD fMRI signal. This work is important because CO₂ is fundamental to cerebral blood flow regulation yet through mechanisms that are somewhat unclear and/or poorly contextualized within the physiology of the system. CO₂ was historically not thought to be involved in functional hyperemia, instead perhaps playing a role constantly for basal blood flow regulation, or via systemic increases in blood CO₂, which is well appreciated to dilate large cerebral blood vessels. The main idea is saturate the 'local CO₂ control system' by saturating the system with systemic CO₂. While ceiling effects are the biggest concern here, the authors attempt to control for this with systemically administered vasoconstrictors. The baseline shifts in CBF are shown upfront in the data, which is strength of the study. Using different techniques (direct O₂ sensors) the authors then explore a potential mechanisms of CO₂ hyperemia via the sodium bicarbonate co-transporter in astrocytes using cre-lox knockout. This work is also important because exactly how astrocyte contribute to cerebral blood flow regulation is unclear, especially regarding Ca²⁺-dependent mechanisms. This work potentially describes a new, maybe Ca²⁺ independent, form of CBF control via astrocytes. All the data are clearly and beautifully presented. While overall an exciting concept and study, I have a number of significant concerns with the some of the general approaches to target this question, how it fits into existing literature on this topic, as well as the data interpretation.

Response: We would like to thank this referee for his/her time taken to review our paper and overall positive assessment of our work. Over 130 years ago Roy and Sherrington proposed that the 'chemical products of cerebral metabolism' trigger cerebrovascular dilations' (PMID: 16991945). Since then numerous studies have attempted to elucidate the signaling mechanisms of the neurovascular response focusing mainly on neurotransmitter release and astroglial Ca²⁺, while the originally proposed metabolic signals have largely been ignored and/or their role not addressed experimentally. A current view of the brain functional hyperaemia driven by Ca²⁺-dependent release of vasoactive substances by astrocytes had been challenged by the results of several studies, in particular by the evidence that activity-dependent cerebral arteriole dilations often precede (e.g. PMID: 23658179) and/or may occur in the absence (e.g. PMID: 26126870) of astrocytic Ca²⁺ signals. To the best of our knowledge our study provides the first experimental data in support of the oldest, but largely discounted, hypothesis of the mechanism that links neuronal activity with local cerebral blood flow via the actions of CO₂. Please review our detailed responses to all the comments raised.

Critique: Controlling for baseline blood flow with systemic indo and caffeine is not very selective. Indo blocks COX but also blocks PKA (PMID: 214715), and can directly affect functional hyperemia too (more below), Caffeine has numerous targets on the adenosine system and on Ca²⁺ stores in many cell types. Indo blocks CO₂ reactivity (PMID: 643149) and both drugs would have systemic effects on blood pressure too.

Response: We thank the reviewer for this comment and agree that combined treatment with indomethacin and caffeine may not be very selective. However, as the effects of CO₂ on cerebrovasculature are so potent, we found that by only using this treatment we could effectively reverse the effect of systemic hypercapnia on global cerebral blood flow. The actions of neither caffeine nor indomethacin given alone were able to fully offset the effect of CO₂ and reduce CBF to the baseline level (Supplementary Figure 5). Importantly, we found that across all the experimental conditions of this study, only ~30% (on average) of the neurovascular response in the somatosensory cortex was reduced by indomethacin and caffeine given individually or in combination. Please review the revised text on Page 7, revised Figure 3 and Supplementary Figure 5.

While preparing the draft of this rebuttal letter we reviewed the results of several published studies that aimed to investigate the mechanisms underlying the effects of CO₂ on cerebral vessels, focusing on the effects of indomethacin and caffeine. We found studies that reported complete blockade of cerebrovascular CO₂ reactivity with indomethacin (as pointed by the reviewer), but also a significant number of studies that showed no effect of indomethacin as well as of other cyclooxygenase inhibitors. Considering a significant number of studies reporting conflicting data, we concluded that the most unbiased approach to answer the comments of the Referee would be to undertake a systematic review of the literature data. After reviewing hundreds of research reports, 131 primary sources reporting the data obtained under 214 different experimental conditions were selected as they met our selection criteria for inclusion in meta-analysis. This analysis is now included in the revised manuscript (revised Figure 1; Supplementary Table 1 and Supplementary References). We considered that inclusion of meta-analysis data in this paper would be appropriate because of the following reasons: 1) the results of the meta-analysis support the main hypothesis of the study; 2) the data describing the effects of cyclooxygenase and adenosine receptor blockade on cerebrovascular response to CO₂ are summarized, and can be used to evaluate the appropriateness of the pharmacological approach used in the study; and 3) the results of meta-analysis provide a valuable resource to the researchers working in the field of cerebral blood flow control.

We are aware of non-specific effects of indomethacin, in particular in blocking the activity of PKA (in addition to inhibition of cyclooxygenase activity). Therefore, the effects of indomethacin were analyzed and reported separately from that of other cyclooxygenase inhibitors. Please review the Figure 1 of the revised manuscript. We report that indomethacin reduces CO₂ effects on cerebrovasculature by 44% (average of 48 studies), while the other cyclooxygenase inhibitors reduce CO₂ evoked responses by a mere 7% (average of 12 studies). Thus, it appears that the ability of indomethacin to block PKA activity (and/or other targets) is indeed relevant in this context, yet indomethacin reduces the cerebrovascular response to CO₂ only partially.

We identified 5 studies that used caffeine to study the effects of adenosine receptor blockade on CO₂-induced cerebrovascular response. Caffeine was found to reduce CO₂-induced response by 26% (5 studies), and this effect (on average) was not different from that of other adenosine receptor blockers (e.g. theophylline).

We now discuss these issues in the revised manuscript and provide data illustrating the effects of combined treatment with indomethacin and caffeine on blood pressure and heart rate recorded in our experiments (Supplementary Figure 4).

Critique: More specific concerns are below, but here, it would have been much more elegant if the authors selectively controlled for tone in the forepaw region using vascular DREADDs previously delivered using locally injected AAV, followed by systemic administration of CNO during the imaging experiment. Perhaps via mural cell expression of Gq or Gi DREADD to decrease blood flow to the level they need. While this would have taken the generation of a custom virus and some validation, I don't think this technique should be out of reach, and may be necessary to achieve clear conclusions about the results.

Response: We agree and have significant expertise in viral gene transfer *in vivo* (e.g. PMID: 25834053). However, in our opinion a significant reduction in cerebral blood flow across a large area of somatosensory cortex would be difficult to achieve using this approach. Although, successful transfection of cerebrovascular endothelial cells to express DREADDGq was reported (PMID: 30482689), to the best of our knowledge targeting vascular smooth muscle cells or capillary pericytes to express DREADDs has not yet been achieved. Moreover, it is not clear which cells to target; arteriole smooth muscle cells, pericytes, or both. It is likely that supplying pial vessels would also need to be targeted, as constriction of smaller downstream vessels may not be sufficient to reduce local blood flow to the baseline level, especially in conditions of CO₂-induced global increases in cerebral perfusion. Finally, CNO has off-target effects (PMID: 28774929; PMID: 29497149) and its use could also be criticized for the lack of specificity.

Critique: Indo and caffeine effects are hard to interpret. Looking at the data, even though they restore baseline blood flow, they appear to block functional hyperemia, at least in the high CO₂ condition. More specifically, in 10% CO₂, using stronger stimulation (3mA) there is a BOLD response, but in the presence of indo and caffeine, still in 10% CO₂, baseline has decreased but 3mA stim produces no BOLD response. This suggests indo and caffeine are not just controlling for baseline blood flow, but also indirectly block functional hyperemia in this high CO₂ state. It is appreciated that indomethacin reduces blood flow increases to systemic hypercapnia, which is likely why baseline blood flow comes down, yet there could be additional effects of indo and caffeine on functional hyperemia itself only in this high CO₂ state, unrelated to the presumed CO₂-mediated neurovascular coupling. How do the authors explain this loss of the 3mA stim induced BOLD signal in indo and caffeine?

Response: We thank the reviewer for this comment. Please review the CBF data shown on Figure 3 of the revised manuscript (Figure 2 of the original submission). In our experimental conditions, 10% CO₂ largely blocks the neurovascular response induced by 1.5 mA stimulation (reduction by 83%), but CO₂ actions also inhibit the evoked neuronal activity (Figure 3f). To compensate for CO₂-induced inhibition, we increased the intensity of stimulation 2-fold; 3 mA stimulation triggered stronger neuronal activations (spike power was 25% higher compared to that of 1.5 mA in normocapnia), yet the neurovascular response remained suppressed by >50% in conditions of 10% inspired CO₂. The reviewer correctly noted that treatment with indomethacin and caffeine in conditions of 10% CO₂ further reduced the response to 3 mA stimulation which remained in 10% CO₂ and completely abolished the response to 1.5 mA stimulation. Importantly, withdrawal of inspired CO₂ in conditions of continuing presence of systemic indomethacin and caffeine restored ~50% of the initial CBF response and led to a full recovery of the BOLD response (Figure 3).

We thank the reviewer for raising this comment and now include the following text in the revised manuscript:

"Analysis of the effects of combined treatment with caffeine and indomethacin across different experimental conditions revealed 11%, 29% and 51% reduction in the magnitude of the CBF increase in the S1FL induced by 1.5 mA in 10% CO₂; 3 mA in 10% CO₂ and 1.5 mA in normocapnia, respectively (control response evoked by 1.5 mA stimulation in normocapnia was taken as 100%). Thus, in our experimental conditions systemic caffeine and indomethacin treatment was found to reduce the neurovascular response by ~30% on average. This result is fully consistent with the data reported in the literature⁵, suggesting that the signaling pathways sensitive to blockade by caffeine and indomethacin partially contribute to the development of the neurovascular response, however the bulk of the response is mediated by a separate (COX- and adenosine receptor-independent) mechanism, the operation of which is fully blocked by a surplus of exogenous CO₂".

Critique: Or perhaps the indo + caffeine block I highlighted above relates to a direct effect on CO₂ NVC and there is previous literature here that the authors do not adequately address. Indo should block COX1 and a previous report attempted to connect locally produced CO₂ induced vasodilation via COX1 activation in astrocytes. That COX1 is involved in cerebral vasodilation to systemic hypercapnic was clearly shown in a classic general knockout study (PMID: 11282894), though COX-1 here is not involved in sensory induced functional hyperemia. More recently, the astrocyte COX1 pathway was shown to be recruited in response to locally applied CO₂, which elevated astrocyte Ca²⁺ to cause a PG dependent dilation (PMID: 28137973). These previous studies should be at least be mentioned or discussed if space permits.

Response: We agree. Please see our response to the previous comment. By including the results of meta-analysis (Figure 1) we now provide a comprehensive summary of all the preceding literature that describe the potential mechanisms underlying the effects of CO₂ on cerebral vasculature. Indeed, pharmacological or genetic blockade of COX-1 was found to reduce the cerebrovascular CO₂ responses by 56% (average of 4 studies). Thus, the effect of specific COX-1 blockade appears to be stronger(!) than that of indomethacin (reduction by 44%, 48 studies), and substantially stronger than that of other non-specific COX inhibitors (reduction by 7%, 12 studies). The reasons underlying these differences are unclear, but it is important to note that all the studies with specific COX-1 blockade were conducted in anaesthetized experimental animals, while a significant proportion of studies that used a non-specific cyclooxygenase inhibitor were performed in human participants.

Regarding the data on astrocyte Ca²⁺ causing PGE₂-mediated dilations in response to CO₂ (PMID: 28137973): respectfully, we were unable to replicate the principle finding of that study that hypercapnia induces Ca²⁺ signals in cortical astrocytes. The authors acknowledged in the paper that the responses to CO₂ are unreliable in acute brain slices, but in several dedicated trials we were also not able to record reliable cortical astroglial Ca²⁺ signals in response to CO₂ *in vivo* (somatosensory cortex). The authors of PMID 28137973 confirm the observations of Niwa et al 2001 (PMID: 11282894) that COX-1 blockade reduces CO₂-induced cerebrovascular dilations and this result (~60% reduction) is included as one of the datapoints in our meta-analysis data summary (Figure 1).

Critique: It is unclear if acetazolamide would restore functional hyperemia in hypercapnia or not because baseline is very high in acetylzolamide alone, which is raised even further in 5% and 10% CO₂. Perhaps a ceiling effect occurs in this condition? The authors did not constrict the vessels in acetylzolamide to restore baseline and then test forepaw stimulation.

Response: The purpose of the acetazolamide experiment was to determine whether the reduction of the neurovascular response during hypercapnia is due to acidification of the brain tissue. To answer this question, we choose to manipulate the brain extracellular pH using acetazolamide given systemically. The most important result from this experiment is that acetazolamide-induced extracellular acidification in the somatosensory cortex (measured and reported, see revised Figure 2g) was similar in magnitude to acidification induced by 5% inspired CO₂, yet the neuronal, CBF and BOLD responses induced by somatosensory stimulation were not affected (revised Figure 2, Supplementary Figure 1). Moreover, the inhibitory effect of CO₂ on the expression of the neurovascular response was not affected by acetazolamide. These data suggest that inhibition of the neurovascular response by provision of exogenous CO₂ is not due to acidification of the brain tissue. Another important conclusion from this series of experiments is that inhibition of the neurovascular response by added CO₂ is unlikely to be due to a ceiling effect. Please review the CBF data illustrated by Figure 2c: In conditions of systemic acetazolamide action, 10% inspired CO₂ increased the CBF to ~280 ml 100 g⁻¹ min⁻¹, likely revealing the cerebrovascular reserve in this experimental model. CO₂ given in the inspired air in concentration 5% and 10% increased the basal CBF to ~130 and 210 ml 100 g⁻¹ min⁻¹, respectively, while the peak response to somatosensory stimulation (i.e. difference between the baseline CBF and peak increase in CBF) recorded in control conditions (normocapnia) was 39±5 ml 100 g⁻¹ min⁻¹ (n=22) (Figure 2c,e; 3c,d). As the sum of these values (210 ml 100 g⁻¹ min⁻¹ peak CBF at 10% CO₂ + 39 ml 100 g⁻¹ min⁻¹ peak neurovascular response) is lower than the CBF recorded at hypercapnia following systemic acetazolamide (280 ml 100 g⁻¹ min⁻¹), the inhibition of the neurovascular response by surplus of exogenous CO₂ cannot be simply explained by the exhaustion of the cerebrovascular reserve. We thank the reviewer for raising this comment and admit that this issue could have been discussed more thoroughly in our original submission. Please review our revised text (Page 6, last paragraph).

Critique: It is a little unexpected that indomethacin doesn't block functional hyperemia in normocapnia. There are several papers that show this (PMID: 17655958; PMID: 24262992)

Response: Please see our response to one of the earlier comments. We apologize for not addressing the effects of indomethacin on the neurovascular response in our original submission. As we discuss in detail above, in our experimental conditions systemic caffeine and indomethacin treatment was found to reduce the neurovascular response by ~30% on average. The magnitude of this effect is similar to that (reduction by ~25%) reported in one of the papers mentioned by the reviewer (PMID: 24262992). Please note that this effect is also within the range of effects observed in other studies that targeted COX or adenosine receptors individually (PMID: 30481531).

Critique: The use of ASL for baseline and stim induced BOLD fMRI is, on one hand commendable by demonstrating a genuine change in O₂ in the forepaw region, but on the other hand, tricky to place in context of previous work. Though I am not an expert here, MR work typically necessitates long stimulation paradigms to capture the data (here 1.5 min of sensory stimulation). Unfortunately, the temporal resolution usually employed in optical imaging studies of functional hyperemia is much faster. Functional hyperemia, driven by neural mediators, can be up and down in several seconds, thus it would be more interesting to see the effects of CO₂ on this time scale. Maybe locally produced CO₂ is important on the time scale of several tens of seconds, but what about in the first few seconds? Can the authors comment on this?

Response: We agree and attempted to address this issue by using fast scan cyclic voltammetry (FCV) in mice with genetic deficiency of NBCe1. FCV provides sub-second temporal resolution and the results of these experiments are discussed in detail below. In a series of validation studies, we demonstrated a very good correlation between signals recorded using FCV and fMRI (please review Fig 4 in PMID: 32685919).

The use of ASL in the experiments involving manipulations of the level of inspired CO₂ was dictated by the need of precise monitoring and control of the effects of CO₂ on cerebral blood flow. The importance of controlling for changes in perfusion is highlighted by the Reviewer and discussed in detail above. Unlike BOLD fMRI, ASL provides both an absolute measurement of CBF and, tied to this, the ability to reliably capture changes in brain perfusion. In this study the use of ASL was essential to quantify, and disentangle, the effects of somatosensory stimulation convolved with slow changes to baseline perfusion induced by gas challenges and/or pharmacological treatments. However, as the reviewer points out, the use of ASL limits the temporal resolution to a few seconds meaning that relatively long sensory stimuli were required (PMID: 22966219).

Classical experiments by Logothetis and colleagues involving simultaneous recordings of neuronal activity and BOLD signals demonstrated that the neurovascular response develops with a significant delay and outlasts the period of increased neuronal activity. Please review the figure below taken from PMID 11449264 (LFP- local field potentials; MUA – multiunit activity). In our opinion, these profiles of the neurovascular responses to short and long stimuli do not suggest the existence of distinct mechanisms that mediate the early vs the late phase of the response, but to answer this question a separate dedicated study would be required.

Figure 3 Simultaneous neural and haemodynamic recordings from a cortical site showing transient neural response. **a–c**, Responses to a pulse stimulus of 24, 12 and 4 s. Both single- and multi-unit responses adapt a couple of seconds after stimulus onset, with LFP remaining the only signal correlated with the BOLD response. SDF, spike-density function (see text); ePts, electrode ROI—positive time series.

Critique: I really like the idea to knockout NBCe1 in astrocytes as a mechanistic link between CO₂ and vasodilation. This is a novel and exciting experiment. However, the authors did not setup this experiment properly: there is no control for tamoxifen. They need a third group that receives tamoxifen in the same animals that are negative for cre or the floxed gene. Otherwise it is impossible to know whether the effect they report is caused by tamoxifen itself, rather than gene knockdown.

Response: We completely agree. The data obtained in a group of animals negative for cre and treated with tamoxifen are now included in the revised submission (revised Figure 4).

Critique: What is baseline CBF like in the eNBCe1 knockout? This is equally important as the other data. If baseline is up significantly, then baseline needs to be controlled for to ensure there is no ceiling effect.

Response: We appreciate that lack of absolute measurements of resting cerebral blood flow is a limitation of this mouse experiment. High resolution fMRI scanning equipment was not available at the Centro de Estudios Científicos in Valdivia (Chile) where the experiments in NBCe1 knockout mice were performed. To answer this question of the reviewer (albeit indirectly) we recorded ventilatory responses to CO₂ in mice with conditional deletion of NBCe1 in astrocytes. As central respiratory sensitivity to CO₂ is dependent on cerebral perfusion (PMID: 16931556), ventilatory responses to CO₂ would be expected to be reduced if deletion of NBCe1 in astrocytes is associated with a sustained increase in global cerebral blood flow. However, the resting ventilation and CO₂-induced increases in the respiratory rate and tidal volume were found to be similar in two groups of animals (Supplementary Figure 6), suggesting that cerebral blood flow is not significantly affected by NBCe1 deficiency.

In the revised submission we now include additional experimental data which support the conclusions of the study. Figure 4d,e provides summary data illustrating the behaviour of animals from 3 experimental groups in an open field. The data suggest that conditional deletion of NBCe1 in astrocytes is associated with increased anxiety/hyperactivity. As CO₂ is a potent trigger of anxiety behaviour (e.g. PMID: 27598969), these data are consistent with the hypothesis that NBCe1 deficiency impairs cellular mechanisms of brain CO₂ clearance.

Critique: Why is the NBCe1 knockout work not done using ASL MR like the rest of the study? Instead the authors use local oxygen partial pressure measurements. It would be nice to see correspondence in the data between the different experiments using a single measurement.

Response: Please review our response to one of the previous comments, but lack of appropriate equipment was not the main reason why the neurovascular responses in mice were assessed via direct measurements of brain tissue PO₂ instead. Experiments involving saturation of the brain CO₂-sensitive vasodilatory mechanism with exogenous CO₂ (illustrated by revised Figures 3 and 4) were conducted in rats and involved analysis of the neurovascular coupling responses using ASL and BOLD fMRI. ASL fMRI can provide an accurate measure of resting CBF in mice, but in our experience can yield high variability when assessing the functional hyperemic response to sensory stimuli in a small region of the brain. As we mention above, in our validation studies conducted in rats we demonstrated a very good correlation between signals recorded using FCV and fMRI (PMID: 32685919).

Critique: The immuno images for NBCe1 do not look comparable to each other, and they also do not look convincing that the authors have detected the protein of interest on astrocytes and they have successfully knocked it down. A more convincing immuno-fluorescence or biochemical approach must be shown for this knockdown experiment.

Response: We thank the reviewer for this comment and apologize for not providing a clear referencing to our earlier study that reported the development and characterization of this animal model (PMID: 33033238). The genetic approach we used results in a partial NBCe1 knockdown (as assessed at the RNA and protein level) and mosaic pattern of NBCe1 expression, which is typical for the experiments of this type. In the revised manuscript we now provide higher magnification images of expression (Figure 4c) and give a more detailed description of the model in the text of the manuscript.

Reviewer #2

Hosford et al. investigate the role of CO₂ in neurovascular coupling by saturating brain CO₂-sensing mechanisms with inspired exogenous CO₂, reasoning that this will nullify responses of blood vessels to further increases in CO₂ that result from neuronal activation and increased metabolic activity. Using this paradigm, they find that a CO₂ is a major contributor to cerebral blood flow responses to neuronal activity, independent of tissue pH. The paper is well written, the data are quite compelling and the main effect of CO₂ on CBF/BOLD responses is striking, and argue for the integration of this mechanism into our current understanding of neurovascular coupling mechanisms. I have the following questions/concerns at this point.

Response: We thank Dr Longden for his time taken to review our paper and overall positive assessment of our work. Below we provide our detailed responses to all the critical comments raised.

Critique: In order to compensate for the possibility that changes in basal CBF are confounding the interpretation of the data by lowering the vasodilatory reserve of vessels, the authors utilize caffeine and indomethacin to return CO₂-induced elevations in CBF back to baseline levels. The data indicate that under these conditions, CO₂ still eliminates the CBF response to forepaw stimulation. Following this, a return to normocapnia leads to normal CBF/BOLD responses, suggesting that the pathways blocked by indomethacin and caffeine do not participate in neurovascular coupling. However, this finding contradicts a number of studies that have found a substantial contribution of the cyclooxygenase pathway, targeted by indomethacin, to neurovascular coupling in both rodents and humans in a range of preparations (see references below). Why is no contribution of these pathways observed under the present conditions?

Takano T, Tian GF, Peng W, Lou N, Libionka W, Han X, Nedergaard M. Astrocyte-mediated control of cerebral blood flow. *Nat Neurosci*. 2006 Feb;9(2):260-7. doi: 10.1038/nn1623. Epub 2005 Dec 25. PMID: 16388306.

Lecrux C, Toussay X, Kocharyan A, et al. Pyramidal neurons are "neurogenic hubs" in the neurovascular coupling response to whisker stimulation. *J Neurosci*. 2011;31(27):9836-9847. doi:10.1523/JNEUROSCI.4943-10.2011

Lacroix A, Toussay X, Anenberg E, Lecrux C, Ferreirós N, Karagiannis A, Plaisier F, Chausson P, Jarlier F, Burgess SA, Hillman EM, Tegeder I, Murphy TH, Hamel E, Cauli B. COX-2-Derived Prostaglandin E₂ Produced by Pyramidal Neurons Contributes to Neurovascular Coupling in the Rodent Cerebral Cortex. *J Neurosci*. 2015 Aug 26;35(34):11791-810. doi: 10.1523/JNEUROSCI.0651-15.2015. PMID: 26311764; PMCID: PMC6705452.

Response: We thank Dr Longden for raising this comment which is similar to one of the criticisms raised by the first Reviewer. Our meta-analysis of published data reporting the effects of pharmacological or genetic blockade of all hypothesized signaling mechanisms indicated that cyclooxygenase pathway contributes, in part, to the development of the neurovascular coupling response (please review Fig 2 in PMID: 30481531). For this revision we conducted a similar systematic review and analysis of published data reporting the effects of pharmacological or genetic blockade of signaling mechanisms that were proposed to mediate the effects of CO₂ on cerebral vasculature (revised Figure 1). This analysis shows that indomethacin has a much stronger effect on cerebrovascular CO₂ reactivity, when compared to other non-specific cyclooxygenase inhibitors, suggesting that indomethacin has off-target effects (e.g. it blocks PKA activity as pointed by Reviewer 1; PMID: 214715). In the present study we applied indomethacin systemically at 10 mg/kg that was sufficient to offset the effect of CO₂ on cerebral blood flow, but only when it was applied in combination with caffeine. The actions of indomethacin given alone in this dose were not sufficient to reduce the CBF to the baseline level. In our opinion these data are not directly comparable to that reported in the preceding literature as in the studies quoted by the reviewer indomethacin was applied either topically or intracisternally and in high concentrations. For example, in the study of Takano et al. indomethacin was applied in a concentration of 500 μM, while IC₅₀ of the drug in blocking COX-1 is in low nM range (20-60 nM). The other two studies quoted by the reviewer do not state the dose of indomethacin used.

We did not discuss this point in our original submission, but Reviewer #1 noted a partial reduction of the neurovascular response following administration of indomethacin and caffeine in our experiments and we now addressed their comment by further analysis of the obtained data. In the revised manuscript we now address the comments of Reviewer 1 and Dr Longden by providing the following additional description of the data:

"Analysis of the effects of combined treatment with caffeine and indomethacin across different experimental conditions revealed 11%, 29% and 51% reduction in the magnitude of the CBF increase in the S1FL induced by 1.5 mA in 10% CO₂; 3 mA in 10% CO₂ and 1.5 mA in normocapnia, respectively (control response evoked by 1.5 mA stimulation in normocapnia was taken as 100%). Thus, in our experimental conditions systemic caffeine and indomethacin treatment was found to reduce the neurovascular response by ~30% on average. This result is fully consistent with the data reported in the literature⁵, suggesting that the signaling pathways sensitive to blockade by caffeine and indomethacin partially contribute to the development of the neurovascular response, however the bulk of the response is mediated by a separate (COX- and adenosine receptor-independent) mechanism, the operation of which is fully blocked by a surplus of exogenous CO₂".

Critique: The authors do not note the timing of this experiment in the methods, i.e. the duration of the rest periods between the different intensities of electrical forepaw stimulation, and the length of time that was allowed for the washout of inhaled CO₂ upon the return to normocapnia. How long were these periods? Could a long wait have led to the metabolism of indomethacin to ineffective concentrations? What is the effect of these compounds on the CBF/BOLD response under normocapnic conditions immediately after administration of indomethacin/caffeine, when they are at their maximal plasma concentrations?

Response: Thank you for this comment and we apologize for not providing this important information in our original submission. The total duration of the experimental protocol was less than 90 min; the last episode of somatosensory stimulation in conditions of continuing systemic caffeine/indomethacin action and following the CO₂ withdrawal was given 30-45 min after the administration of the drugs. Therefore, the absence of a strong indomethacin

effect on the neurovascular response in conditions of normocapnia could not be explained by the washout of the drug, considering that half-life of indomethacin in the rat is between 8-22 hr (PMID: 3104022). We now provide this additional information on the experimental design in the revised manuscript.

Critique: Since these compounds may block pathways that are involved in neurovascular coupling, I suggest that a more 'neutral' drug choice for baseline correction may make data interpretation more straight forward. With this in mind, what is the effect of hypercapnia on CBF/BOLD responses when blood flow is instead corrected using phenylephrine (see Ogoh et al. 2011 below)?

Ogoh S, Sato K, Fisher JP, Seifert T, Overgaard M, Secher NH. The effect of phenylephrine on arterial and venous cerebral blood flow in healthy subjects. *Clin Physiol Funct Imaging*. 2011 Nov;31(6):445-51. doi: 10.1111/j.1475-097X.2011.01040.x. Epub 2011 Jul 12. PMID: 21981455.

Response: We thank the reviewer for this suggestion but disagree that the use of phenylephrine will provide a better solution for controlling cerebral blood flow during hypercapnia. We routinely use challenges of this type to test the sensitivity of the baroreflex. Since the effects of phenylephrine are transient, this experiment would require continuous infusion, which would lead to sustained significant increases in systemic arterial blood pressure, baroreflex activation, heart rate changes, etc. which will affect cerebral blood flow and, therefore, may also have an effect on the expression of the neurovascular response.

Critique: The corollary of this is whether the authors can still reliably increase blood flow well above the already elevated level under 10% CO₂? This could be addressed by administering an L-type voltage-gated calcium channel blocker in the presence of CO₂, for example, which should still produce a robust increase in blood flow.

Response: We agree and, in our opinion, addressed the issue of a potential 'ceiling effect' by the experiment involving systemic treatment with acetazolamide. The main purpose of the acetazolamide experiment was to determine whether the reduction of the neurovascular response during hypercapnia is due to acidification of the brain tissue derived from CO₂ hydration. The most important result from this experiment (revised Figure 2) is that acetazolamide-induced extracellular acidification in the somatosensory cortex was similar in magnitude to that induced by 5% inspired CO₂, yet the neuronal, CBF and BOLD responses induced by somatosensory stimulation were not affected. But another important conclusion that can be derived from this series of experiments is that inhibition of the neurovascular response by inspired CO₂ is unlikely to be due to a 'ceiling effect'.

Please review the CBF data illustrated by Figure 2c: In conditions of systemic acetazolamide action, 10% inspired CO₂ increased the CBF to ~280 ml 100 g⁻¹ min⁻¹, likely revealing the cerebrovascular reserve in this experimental model. CO₂ given in the inspired air in concentration 5% and 10% increased the basal CBF to ~130 and 210 ml 100 g⁻¹ min⁻¹, respectively, while the peak response to somatosensory stimulation (i.e. difference between the baseline CBF and peak increase in CBF) recorded in control conditions (normocapnia) was 39±5 ml 100 g⁻¹ min⁻¹ (n=22) (Figure 2c,e; 3c,d). As the sum of these values (210 ml 100 g⁻¹ min⁻¹ peak CBF at 10% CO₂ + 39 ml 100 g⁻¹ min⁻¹ peak neurovascular response) is lower than the CBF recorded at hypercapnia following systemic acetazolamide (280 ml 100 g⁻¹ min⁻¹), the inhibition of the neurovascular response by surplus of exogenous CO₂ cannot be simply explained by the exhaustion of the cerebrovascular reserve. We thank the reviewer for raising this comment and admit that this issue could have been discussed more thoroughly in our original submission. Please review our revised text (Page 6, last paragraph).

Critique: A number of studies in the past have used 5% CO₂/95% O₂, or 5% CO₂/20% O₂/balance N₂ to gas solutions which are used to perfuse brain slices and other preparations to study neurovascular coupling. It seems that, according to the present data, these conditions should greatly blunt neurovascular coupling/vasodilations, yet these are still readily observable under these conditions. How should the data from the present study be integrated with these findings?

Response: Thank you. Gas mixtures containing 5% CO₂ are indeed routinely used in the majority of studies conducted in brain slices. This is dictated by the need of mimicking the physiological environment as in the presence of 26 mM HCO₃⁻ (concentration of bicarbonate usually used in aCSF solutions) saturation of aCSF with 5% CO₂ results in a pH of 7.4 and partial pressure of CO₂ ~40 mmHg – similar to that of the arterial blood. To study the effects of CO₂ *in vitro* much higher concentrations are needed. In our experience, in cortical slices application of solutions saturated with 10-14% CO₂ is required to induce reliable dilations of cerebral arterioles. Thus, cerebrovascular CO₂ sensitivity appears to be significantly reduced *in vitro*.

Critique: Neurovascular coupling is characterized as resulting from communication between neurons and astrocytes with vascular smooth muscle cells and pericytes, and the contribution of the endothelium appears to have been overlooked. Our group also recently provided direct evidence that parenchymal cells communicate with the endothelial cells that compose the capillaries.

Longden TA, Dabertrand F, Koide M, et al. Capillary K⁺-sensing initiates retrograde hyperpolarization to increase local cerebral blood flow. *Nat Neurosci.* 2017;20(5):717-726. doi:10.1038/nn.4533

Response: Thank you. We are fully aware of this excellent work and mentioned the role of K⁺ in mediating the neurovascular coupling response in our original submission. Please see our response to the next comment on how the data supporting the existence of multiple parallel, sequential and interacting pathways can be integrated into a unifying model of neurovascular coupling.

Critique: The data indicate that CO₂ makes a major contribution to neurovascular coupling and that with CO₂-sensing mechanisms saturated, neurovascular coupling does not proceed. Emerging evidence from a number of labs supports important contributions of a range of mediators to this process, including nitric oxide, K⁺, prostaglandins, and epoxyeicosatrienoic acids, among others, as the authors acknowledge. According to the data herein, it appears that CO₂ sensing thus underpins all (or a very large fraction) of these diverse mechanisms. It is important to discuss how, at a mechanistic level, CO₂ sensing might underlie these, as this is not clarified in detail in the present version of the manuscript. The authors do note that ATP release may be a CO₂-sensitive process, and this discussion point could be expanded to provide a more global view of CO₂'s contribution to other neurovascular coupling mechanisms as well.

Response: We completely agree with this comment. When we entered this research field a few years ago, we conducted a systematic review and meta-analysis of literature data in order to evaluate the relative significance of all hypothesized signaling pathways of neurovascular coupling (PMID: 30481531). Our analysis suggested that all the proposed mechanisms can account for up to ~60% of the neurovascular response, with nitric oxide signaling likely to play a major role. Our analysis also pointed to the existence of an as yet

unidentified signaling mechanism (or mechanisms) responsible for a significant proportion (at least one third) of the neurovascular response.

Motivated by the Reviewers' comments received in this round, we undertook a similar analysis of the literature data that describe the signaling mechanisms underlying cerebrovascular response to CO₂. The data retrieved from 131 sources are now summarized by Figure 1 of the revised manuscript. Supporting our hypothesis, the results of this meta-analysis show that the same key signaling mechanisms that mediate the neurovascular coupling response (PMID: 30481531), also mediate the effects of CO₂ on cerebral blood flow, including nitric oxide and cyclooxygenase pathways. Considering that most of glucose metabolised by the brain tissue ends up as CO₂ in neurons and CO₂ production is proportional to neuronal energy use, it is logical to conclude that CO₂ and increased neuronal activity recruit the same mechanisms leading to the dilation of the parenchymal cerebral vessels. Yet, despite many studies that addressed the mechanisms of cerebrovascular CO₂ sensitivity it remains unclear how exactly CO₂ (or H⁺) is sensed. To address this comment of the reviewer we now provide a more detailed discussion of this issue in the revised manuscript (Page 9).

Reviewer #3

The past 10 years have provided numerous studies showing quite compelling evidence that neuronal activity causes regulation of blood flow via the release of vasoactive molecules from either astrocytes or pericytes. These signals include prostaglandins, 20-HETE, ATP and K⁺ in a process called functional hyperemia. This phenomenon is underlying the BOLD effect utilized in fMRI studies to interrogate brain function. The current paper challenging this dogma, namely that functional hyperemia involves glia, but rather the authors propose that it is entirely due to local CO₂ changes as a result of oxidative metabolism.

Response: We would like to thank this referee for his/her time taken to review our paper and overall positive assessment of our work. We apologize that our narrative was perhaps not sufficiently clear, and the reviewer got an impression that we aim to challenge the current dogma of the mechanisms underlying the neurovascular coupling. In fact, our study provides the first experimental data in support of the very first, but largely discounted, hypothesis of the mechanism that links neuronal activity with local cerebral blood flow via local metabolic changes, proposed by Roy and Sherrington in 1890 (PMID: 16991945). We aim to incorporate the CO₂-mediated signaling into the existing model(s) of the neurovascular coupling mechanism that includes neurons, astrocytes, vascular smooth muscle cells, pericytes, and vascular endothelium.

Critique: Unlike most prior studies, the authors utilize MRI and the BOLD signal as a functional readout, equating an increase in regional bold signal with vasodilation and a decrease with vasoconstriction. The region of interest is the somatosensory cortex and the stimulus used is electrical forepaw stimulation. The major experimental argument that leads the authors then to propose that CO₂ is the principle vasoregulator is the fact that inhalation of super-physiological concentrations of CO₂, namely 10% leads to a complete loss to changes in blood flow in response to electrical forepaw stimulation. However this level of CO₂ causes maximal vessel dilation and increases cerebral blood flow almost 3 fold. One argument one could make therefore, right of the bat, is that no further vasodilation would be possible for vessels already maximally dilated.

Response: We thank the reviewer for raising this comment. Indeed, we were mindful of the global effect of inspired CO₂ on cerebral blood flow and the potential of the chosen level of hypercapnia to exhaust the cerebrovascular reserve. Therefore, using pharmacological treatment involving a combination of indomethacin and caffeine we aimed to reverse the effect of CO₂ on cerebral vasculature and bring the brain perfusion to the level recorded in normocapnia. The effects of all the experimental treatments on brain extracellular pH, cerebral blood flow and the neuronal activity were recorded, controlled, and reported in the paper.

The data obtained in the experiments involving application of 10% inspired CO₂ following systemic treatment with acetazolamide provide additional evidence suggesting that the inhibition of the neurovascular response by inspired CO₂ is unlikely to be due to a 'ceiling effect'. Please review the CBF data illustrated by Figure 2c: In conditions of systemic acetazolamide action, 10% inspired CO₂ increased the CBF to ~280 ml 100 g⁻¹ min⁻¹, likely revealing the cerebrovascular reserve in this experimental model. CO₂ given in the inspired air in concentration 5% and 10% increased the basal CBF to ~130 and 210 ml 100 g⁻¹ min⁻¹, respectively, while the peak response to somatosensory stimulation (i.e. difference between the baseline CBF and peak increase in CBF) recorded in control conditions (normocapnia) was 39±5 ml 100 g⁻¹ min⁻¹ (n=22) (Figure 2c,e; 3c,d). As the sum of these values (210 ml 100 g⁻¹ min⁻¹ peak CBF at 10% CO₂ + 39 ml 100 g⁻¹ min⁻¹ peak neurovascular response) is lower than the CBF recorded at hypercapnia following systemic acetazolamide (280 ml 100 g⁻¹ min⁻¹), the inhibition of the neurovascular response by surplus of exogenous CO₂ cannot be simply explained by the exhaustion of the cerebrovascular reserve. We thank the reviewer for raising this comment and admit that this issue could have been discussed more thoroughly in our original submission. Please review our revised text (Page 6, last paragraph).

Critique: To counterargue this, the authors then give the animals the vasoconstrictors indomethacin and caffeine in the presence of 10% CO₂ and again see no BOLD response to forepaw stimulation. It is known that release of prostaglandin can dilate pre-constricted blood vessels in brain slices. However, this experiment does not proof that CO₂ release from neurons, bypassing glia cause vasodilation.

Response: We agree that more studies using different experimental approaches would be required to provide further support for the hypothesis of a key role played by CO₂ in mediating the neurovascular response. However, in addition to the data presented in this submission, the hypothesis is supported by other lines of evidence: i) most of the glucose metabolised by brain tissue ends up as CO₂ in neurons and CO₂ production is proportional to neuronal energy use; ii) CO₂ has a very strong dilatory effect on cerebrovasculature; and iii) systematic review and analysis of published data reporting the effects of pharmacological or genetic blockade of CO₂- and neuronal activity-induced cerebrovascular dilations revealed that the same key signaling mechanisms that mediate the neurovascular coupling response (PMID: 30481531), also mediate the effects of CO₂ (revised Figure 1), and involve nitric oxide and cyclooxygenase pathways. Therefore, it is logical to hypothesize that CO₂ and increased neuronal activity recruit the same signaling mechanisms. CO₂ is not bypassing glia but requires glial mechanisms for effective transport to the vasculature and eventual clearance from the brain, as our data obtained in mice with conditional deletion of electrogenic sodium-bicarbonate transporter 1 (NBCe1) suggest (please review revised Figure 4).

Critique: Unfortunately the key experiment to substantiating the author' claim that CO₂ released from neurons is necessary and sufficient to regulate blood flow is missing. It is often called the "block" experiment and would have to block CO₂ release from neurons in

spite of forepaw stimulation. In other words disrupting the signaling molecule should no longer lead to changes in blood flow.

Response: We agree but unfortunately the experiment of this type is impossible to design as CO₂ production is the fundamental process of cell metabolism. Blockade of CO₂ release from neurons would require inhibition of TCA cycle and this would undoubtedly impair the neuronal function. Yet, in our opinion we did perform the “block” experiment, but instead of inhibiting CO₂ production by neurons, we interfered with the transport of CO₂ from neurons to the vasculature by deleting NBCe1 in astrocytes. NBCe1 functions as one of the conduits of CO₂ transport across the membrane (PMID: 33633837) and its deletion in astrocytes was found to completely abolish the neurovascular response in our experiments (revised Figure 4f). In the revised submission we now include additional experimental data suggesting that in this model CO₂ elimination from the brain is indeed impaired (revised Figure 4d,e).

Critique: Also, the approach used in this study does not allow to discern whether changes in blood flow occur at the capillary level, in penetrating arterioles and feeding arteries. The studies that so nicely implicate astrocytes as the purveyor of the signal were done with cellular resolution showing activation of single astrocytes causing time-synchronized vasodilatations. It maybe necessary to repeat these studies in the presence of 10% CO₂ and show that now, astrocytes no longer participate. Might it even be possible that it is the astrocytes that sense the CO₂ and ultimately the response, while ascribed to CO₂ still requires glia as an intermediate?

Response: We agree that studies of the processes that occur at the level of individual cellular elements of the neurovascular unit are required to further support the hypothesis and this work is ongoing. However, in response to this comment of the reviewer we would like to make a point that the role of astroglial Ca²⁺ signaling in the development of the neurovascular coupling response remains controversial. There is evidence that neuronal activity-driven changes in flow are not affected by genetic deletion of astroglial IP₃ receptors (PMID: 23658179; PMID: 23785506; PMID: 25253859) and that the cerebral arteriole dilations often precede (PMID: 23658179) and/or occur in the absence of astrocytic [Ca²⁺]_i changes (PMID: 26126870). The importance of astroglial mechanisms for neurovascular coupling is suggested by the data obtained in mice with conditional knockdown of NBCe1 astrocytes (Figure 4), however, the operation of this mechanism does not appear to rely on changes in intracellular Ca²⁺.

Critique: Changing dogma carries a significant burden of proof. While I find these studies compelling, I am not convinced that they are strong enough yet to substantiate the claim that countless other studies got it wrong.

Response: We agree, but in our opinion the hypothesis and the data presented in our manuscript do not suggest that all the previously proposed models of the mechanisms underlying the neurovascular coupling are incorrect. When we entered this research field, we conducted a systematic review and meta-analysis of the literature in order to get an understanding of the relative significance of all hypothesized signaling pathways of the neurovascular response (PMID: 30481531). Our meta-analysis suggested that all the proposed mechanisms can only account for up to ~60% of the neurovascular response, with nitric oxide likely to play a key role. The analysis pointed to the existence of an as yet unidentified signaling mechanism (or mechanisms) responsible for a significant proportion (at least one third) of the response.

Motivated by the comments provided by all the Reviewers of this submission, we undertook a similar analysis of the literature data that describe potential mechanisms underlying cerebrovascular sensitivity to CO₂. The data retrieved from 131 sources are now illustrated by Figure 1 of the revised manuscript. Supporting our hypothesis, the results of this analysis show that the same key signaling mechanisms that mediate the neurovascular coupling response (PMID: 30481531), also mediate the effects of CO₂ on cerebral blood flow, including nitric oxide and cyclooxygenase pathways. As we argue above, one conclusion that can be derived from these data is that CO₂ and increased neuronal activity recruit the same signaling mechanisms. Yet, despite hundreds of studies that addressed the mechanisms of cerebrovascular CO₂ sensitivity it remains unclear how exactly CO₂ (or H⁺) is sensed and this work is ongoing. To address this comment of the Reviewer we now provide a more detailed discussion of this issue in the revised manuscript (Page 9). Thank you.

REVIEWER COMMENTS

Reviewer #1 (Remarks to the Author):

The authors have addressed the majority of my initial comments satisfactorily. I would like to reiterate that I think this study is important, novel and tackling an age-old question. However, I have two more potentially important points. It was not my intent to bring up new concerns on the original data provided, but these comments could be important for the interpretation of this work.

1) The core approach to test the role of local CO₂ in functional hyperemia is really an occlusion experiment with systemic CO₂, but it is difficult to understand if that is really what is going on. The authors have reasonably addressed that the block of functional hyperemia by systemic CO₂ is not due to a ceiling effect caused by elevated baseline blood flow. However, what if systemic CO₂/pH is affecting some other process that decreases NVC? Whether a general pH shift is properly controlled for is still somewhat unclear but could be easily solved with a statistical test. While it is true that acetazolamide itself (normocapnia) decreases pHe in the brain, and that this treatment did not affect functional hyperemia, the pH decrease is comparable to 5% CO₂, whereas 10% CO₂ creates an even larger decrease in pHe. Thus, a key question is whether the pHe data of acetazolamide (normocapnia) is statistically different from the pHe data of 10% CO₂. If they are not statistically different than I think the authors interpretation still stands. However, if they are different, than a general effect of pH cannot be properly ruled out for 10% CO₂. It seems like pH should not be the cause of reduced NVC for 5% CO₂ (supporting the authors position). I am not necessarily asking the authors to do another experiment with a higher dose of acetazolamide to better mimic the pHe drop to 10% CO₂, but this potential caveat needs to be discussed if the statistics do not go in the author's favour.

2) With the involvement of astrocyte NBCe1, it is puzzling why acetazolamide treatment has no effect? With the results from this experiment the authors state that the mechanism they describe is independent of pH and while this may be true that H⁺ are not directly involved, NBCe1 transports bicarbonate, not CO₂, as far as I understand. As brain CO₂ is generated from TCA metabolism and bicarbonate is being transported into astrocytes, carbonic anhydrase should be working for this conversion, yet acetazolamide does not do anything. How can this be? The authors may be skirting this potential issue by stating that NBCe1 "transports CO₂/HCO₃" but I cannot find evidence that the transporter can transport CO₂, only bicarbonate. I have two ideas: 1) it is possible that acetazolamide is not actually getting into the brain, and only the protons associated with the pH change are. There is some older evidence that suggests it may not cross the BBB (citations below). This might explain why the authors can detect the pH drop, yet acetazolamide has no effect on the CO₂ effect; OR 2) Perhaps CO₂ and H₂O can be converted to HCO₃⁻ and H⁺ without carbonic anhydrase. If this is the case, perhaps the reaction to create HCO₃⁻ from CO₂ is still proceeding in the parenchyma when CA is blocked, however, even if so, I would expect things to be happening slower as CA should at least facilitate this reaction. There is no evidence of slow NVC in the author's trace data.

3) After stating the 11, 29 and 51% reduction of NVC by indo and caffeine in the different conditions, the authors then say "Thus, in our experimental conditions systemic caffeine and indomethacin treatment was found to reduce the neurovascular response by ~30% on average." This number and stating 'on average' is confusing because it is across three different conditions. Perhaps just say '...was found to only partially reduce the neurovascular response.'

4) In Supplementary figure 3, it would nice to actually see the power frequency plots in either all of the different conditions or some key ones, like normocapnia, 10% CO₂ and acetazolamide and even high current stim to see how neural activity has been boosted in this experiment. For the latter, it doesn't have to look identical to control but it is important to data to show.

5) There are two conflicting arguments in the new version of the paper: 1) all the blockers for known cell pathways in neurovascular coupling far from entirely block functional hyperemia, thus there is a significant unexplained pathway that could be CO₂; conflicts with: 2) meta-analysis of the literature

reveals that common NVC blockers also block the effects of CO₂ reactivity; thus perhaps CO₂ is a driver of NVC through known pathways. Which stance would the authors like to take? Or can the wording be refined if these ideas are not mutually exclusive.

6) Cartoon showing Na⁺ and HCO₃⁻ movement in and out of cells but grey circles are used for both species, which is confusing. Perhaps use different colours or draw a little stick molecule for HCO₃.

7) In the new figure 1, what does the triangle mean? The picture legend shows circles, but no triangles and I didn't see them defined in the legend text. Also, it would be helpful if the absolute size of the circles picture legend matched the absolute size of the circles in the figure. Right now, the smallest circle in the figure is much smaller than the smallest circle in the legend, which is a little confusing.

Citations

1) Maren, T. H. 1967. Carbonic anhydrase: chemistry, physiology and inhibition. *Physiol. Rev.* 47:595–781.

2) Roth, L. J., J. C. Schoolar, and C. F. Barlow. 1959. Sulfur-35-labelled acetazolamide in cat brain. *J. Pharm. Exp. Ther.* 125:128–136.

Reviewer #2 (Remarks to the Author):

Thank you for taking the time to address my comments. I have the following minor comments regarding the revised version of the manuscript.

Response 1: The addition of the meta-analysis is a nice touch and the remarks about concentration concerns and off target effects are reasonable, but the additional text is not clear to me - as I understand this you are referring to the data summarized in figure 2d and specifically to the means? This should be stated if so.

Also, the 1.5 mA normocapnia condition is stated to be different but the P value of 0.07 does not indicate that there is a statistically significant difference between these two datasets. The text here should be adjusted to reflect this.

Reviewer #3 (Remarks to the Author):

This paper posits that the signaling underlying neuromuscular coupling is poorly understood and that the role of CO₂ in the process had been unknown.

I recall from Physiology lecture in the 1980s that CO₂ was the principle regulator of blood flow to the brain and underlies auto regulation. Moreover, the Bohr effect (1904) places CO₂ central in the gas exchange between hemoglobin and deoxyhemoglobin.

Hence I and surely many others never questioned but assumed that CO₂ was the principle regulator of blood flow in the brain. I always considered the astrocytic regulation via PGE₂ and HETE an added mechanisms. I obviously stand corrected as this manuscript finally confirms what I had assumed to be the case.

There is an interesting recent publication studying the same question in humans.

Caldwell HG, Howe CA, Hoiland RL, Carr JMJR, Chalifoux CJ, Brown CV, Patrician A, Tremblay JC, Panerai RB, Robinson TG, Minhas JS, Ainslie PN. Alterations in arterial CO₂ rather than pH affect the kinetics of neurovascular coupling in humans. *J Physiol.* 2021 Jun 9. doi: 10.1113/JP281615. Epub ahead of print. PMID: 34107079.

Of course in the present study using rodents the authors were able to examine the the astrocyte specific knockout of the Na bicarbonate exchanger showing that this is sufficient to block neuromuscular coupling. Yet rather than strengthening the conclusion this puzzles me as no mechanistic answer is provided as to how the astrocyte then causes the smooth muscle to relax. I could see how pH would do that but apparently not. So how does CO₂ cause vasodilation? Astrocytes seem to be essential yet how is not shown. Hence, while others may be enthusiastic about showing that CO₂ causes vasodilation and that this will supersede all other neuromuscular coupling, I find no new mechanistic insight in the paper.

I am a bit puzzled by this statement: "that CO₂ diffusion across cellular membranes is facilitated by certain membrane channels and transporters" ...gases move freely across membranes irrespective of channels or transporters.

also it is misleading to refer to "CO₂/HCO₃⁻ transport in astrocytes" as CO₂ is not transported. The transporter exchanges Na and bicarbonate.

Manuscript ID: NCOMMS-20-46461A
Responses to the referees' comments

We would like to thank the reviewers and the Editors for their time taken to evaluate our revised submission and overall positive assessment of our work. We are grateful for further detailed and constructive comments provided and delighted to have an opportunity to re-submit our work. In this rebuttal letter we provide a full response to all the additional criticisms raised by the reviewers and submit the second revision of our manuscript.

Reviewer #1:

The authors have addressed the majority of my initial comments satisfactorily. I would like to reiterate that I think this study is important, novel and tackling an age-old question. However, I have two more potentially important points. It was not my intent to bring up new concerns on the original data provided, but these comments could be important for the interpretation of this work.

Response: We thank this referee for his/her time taken to review our paper and positive assessment of our work. Please review our detailed responses to all the additional comments raised. We found these comments extremely helpful and revised the text and the figures of the manuscript accordingly.

Critique: The core approach to test the role of local CO₂ in functional hyperemia is really an occlusion experiment with systemic CO₂, but it is difficult to understand if that is really what is going on. The authors have reasonably addressed that the block of functional hyperemia by systemic CO₂ is not due to a ceiling effect caused by elevated baseline blood flow. However, what if systemic CO₂/pH is affecting some other process that decreases NVC?

Response: We thank the Reviewer for raising this comment. We carefully considered and attempted to control for changes in three most obvious parameters/processes sensitive to changes in CO₂/pH:

1) Potential ceiling effect caused by CO₂-induced increases in baseline cerebral blood flow. We present and discuss the data suggesting that the inhibition of the neurovascular response by surplus of exogenous CO₂ cannot be simply explained by the exhaustion of the cerebrovascular reserve. Thank you for acknowledging that we satisfactorily addressed this issue in our first rebuttal letter and in the text of the revised manuscript;

2) CO₂ or acidification-induced inhibition of the neuronal activity. As expected, hypercapnia had an inhibitory effect on the evoked neuronal activity in the S1FL region of the cerebral cortex (Fig 3f). In order to control and compensate for the CO₂-induced neuronal inhibition, the intensity of stimulation in hypercapnic conditions was increased to ensure the evoked neuronal activity in the somatosensory cortex is similar between the experimental conditions (at normocapnia and at 10% inspired CO₂, Fig 3f). At the same level of the neuronal activity, the neurovascular response remained to be strongly suppressed by surplus of exogenous CO₂ (Fig 3);

3) Acidification of the brain tissue, as highlighted by the Reviewer and discussed next.

Critique: Whether a general pH shift is properly controlled for is still somewhat unclear but could be easily solved with a statistical test. While it is true that acetazolamide itself (normocapnia) decreases pHe in the brain, and that this treatment did not affect functional

hyperemia, the pH decrease is comparable to 5% CO₂, whereas 10% CO₂ creates an even larger decrease in pHe. Thus, a key question is whether the pHe data of acetazolamide (normocapnia) is statistically different from the pHe data of 10% CO₂. If they are not statistically different than I think the authors interpretation still stands. However, if they are different, than a general effect of pH cannot be properly ruled out for 10% CO₂. It seems like pH should not be the cause of reduced NVC for 5% CO₂ (supporting the authors position). I am not necessarily asking the authors to do another experiment with a higher dose of acetazolamide to better mimic the pHe drop to 10% CO₂, but this potential caveat needs to be discussed if the statistics do not go in the author's favour.

Response: As suggested by the Reviewer, we compared the effect of acetazolamide on brain tissue extracellular pH with that of 5% and 10% inspired CO₂. There was no difference between the effect of 5% CO₂ and acetazolamide on brain tissue extracellular pH, but 10% inspired CO₂ led to a more profound acidification which was statistically significant compared to that induced by acetazolamide. The dose of acetazolamide used in our study (10 mg kg⁻¹, i.v.) was fairly high, therefore, increasing the dose further was not expected to result in a more profound extracellular acidification of the brain tissue. The following lines of evidence support the conclusion that surplus of exogenous CO₂ inhibits the neurovascular response independently of changes in brain tissue pH:

1. Acetazolamide decreased brain tissue pH by the same degree as 5% inspired CO₂. While 5% CO₂ given in the inspired air strongly inhibited the neurovascular response induced by somatosensory stimulation, acetazolamide had no effect (Fig 2c,e,f and Suppl Figs 1 and 3).

2. Acetazolamide also had no effect on the magnitude of the CO₂-induced increases in CBF (Suppl Fig 2). Thus, in conditions of brain tissue acidification induced by systemic acetazolamide treatment both the neuronal activity-induced and CO₂-evoked cerebrovascular responses were preserved. This result is consistent with the main hypothesis of the study that the neurovascular response is mediated by the actions of CO₂.

3. Although the difference in the degree of extracellular acidification induced by acetazolamide and 10% inspired CO₂ was found to be statistically significant, it is questionable if this difference has a biological significance. Acetazolamide treatment led to extracellular acidification by -0.11 ± 0.01 pH units, whereas 10% inspired CO₂ decreased brain tissue pH by -0.19 ± 0.02 pH units – a difference of a mere 0.08 pH units.

We thank the reviewer for raising this important comment and discuss these points in the revised manuscript, as appropriate.

Critique: With the involvement of astrocyte NBCe1, it is puzzling why acetazolamide treatment has no effect? With the results from this experiment the authors state that the mechanism they describe is independent of pH and while this may be true that H⁺ are not directly involved, NBCe1 transports bicarbonate, not CO₂, as far as I understand. As brain CO₂ is generated from TCA metabolism and bicarbonate is being transported into astrocytes, carbonic anhydrase should be working for this conversion, yet acetazolamide does not do anything. How can this be? The authors may be skirting this potential issue by stating that NBCe1 "transports CO₂/HCO₃" but I cannot find evidence that the transporter can transport CO₂, only bicarbonate. I have two ideas: 1) it is possible that acetazolamide is not actually getting into the brain, and only the protons associated with the pH change are. There is some older evidence that suggests it may not cross the BBB (citations below). This might explain why the authors can detect the pH drop, yet acetazolamide has no effect on the CO₂ effect; OR 2) Perhaps CO₂ and H₂O can be converted to HCO₃ and H⁺ without

carbonic anhydrase. If this is the case, perhaps the reaction to create HCO_3^- from CO_2 is still proceeding in the parenchyma when CA is blocked, however, even if so, I would expect things to be happening slower as CA should at least facilitate this reaction. There is no evidence of slow NVC in the author's trace data.

Response: We thank the reviewer for raising this comment and for suggesting several important discussion points. Below we consider each of them individually:

1. Transport of CO_2 by NBCe1: Historically, CO_2 was thought to diffuse through the plasma membrane freely. A significant body of work by Walter Boron and colleagues led to the understanding that the diffusion of CO_2 across the biological membrane is restricted and to a large extent is mediated by certain membrane channels, such as aquaporin water channels. More recent evidence suggests that CO_2 can also diffuse through the NBCe1. In a recent review article Michenkova and colleagues (PMID: 33633837) discuss the results of three different assays that all provided evidence in support of the hypothesis that NBCe1, in the presence of HCO_3^- , has a substantial CO_2 conductance. The authors suggested that either bicarbonate has an allosteric effect on NBCe1, thereby creating a CO_2 channel through the protein and/or NBCe1 cycles through conformational changes during $\text{Na}^+/\text{HCO}_3^-$ cotransport with CO_2 moving through a transiently available CO_2 pathway. Characteristically high expression of NBCe1 in astrocytes can, therefore, be explained by this property of the transporter which together with aquaporin channels are responsible for the effective CO_2 transport across the astroglial membrane.

2. Acetazolamide not getting into the brain: We thank the reviewer for drawing our attention to these earlier works. There is a general consensus that the permeability of the blood brain barrier to H^+ is rather low, therefore, in our opinion the action of acetazolamide on brain tissue pH cannot be explained by blood acidification as a result of the peripheral action of the drug. We recently reported that acetazolamide produces significant localized extracellular acidification when injected directly into the brain tissue (PMID: 33033238).

3. CO_2 and H_2O can be converted to HCO_3^- and H^+ without carbonic anhydrase. This is certainly true. CO_2 and H_2O are converted to HCO_3^- and H^+ without the involvement of carbonic anhydrase at a relatively high speed with a first-order rate constant of 0.15 s^{-1} (10^6 s^{-1} with carbonic anhydrase II). As noted by the reviewer, systemic treatment with acetazolamide had no significant effect on the profile of the cortical CBF responses induced by somatosensory stimulation (or inhaled CO_2) suggesting that the activity of carbonic anhydrase plays no significant role in the development of the neurovascular coupling response.

Collectively the data obtained in animals with conditional NBCe1 knockdown in astrocytes suggest that NBCe1 deficiency impairs the cellular mechanisms of brain CO_2 clearance and are consistent with the hypothesis that NBCe1 acts as a conduit of CO_2 transport across the astroglial membrane. There is also evidence that in the brain molecular CO_2 can be sensed directly (i.e. not via a proxy of associated pH changes) by hemichannels which belong to the beta-connexin family (such as connexin 26, connexin 30, and connexin 32) which increase their open probability (allowing egress of vasoactive molecules) proportionally to the concentration of CO_2 which forms carbamate bridges between the subunits forming the channel (see for example PMID:24220509). If so, enzymatic conversion of $\text{CO}_2/\text{H}_2\text{O}$ to $\text{HCO}_3^-/\text{H}^+$ may not be essential for CO_2 sensing and/or CO_2 transport in the brain. We now discuss these points in the revised manuscript as appropriate. Thank you.

Critique: After stating the 11, 29 and 51% reduction of NVC by indo and caffeine in the different conditions, the authors then say "Thus, in our experimental conditions systemic

caffeine and indomethacin treatment was found to reduce the neurovascular response by ~30% on average." This number and stating 'on average' is confusing because it is across three different conditions. Perhaps just say '...was found to only partially reduce the neurovascular response.'

Response: We thank the Reviewer and agree that averaging the effects of the drugs across different experimental conditions was not entirely appropriate. For this revision we re-analyzed the data and the new analysis revealed that the effect of combined treatment with caffeine and indomethacin on the expression of the neurovascular response was only significant when forepaw stimulation was applied with 3 mA current and in conditions of 10% inspired CO₂. We now revised the text of the manuscript to read:

"As no significant CBF and BOLD responses to somatosensory stimulation were recorded at 10% inspired CO₂ following systemic treatment with caffeine/indomethacin (Figure 3c-e), this pharmacological treatment appeared to have an inhibitory effect on the neurovascular response in hypercapnia. CBF increases in the S1FL region induced by 3 mA stimulation at 10% inspired CO₂ were reduced by 29% (p=0.02) following systemic treatment with caffeine and indomethacin (Figure 3d). This result is consistent with the data reported in the literature⁵, suggesting that the signaling pathways sensitive to blockade by caffeine and indomethacin partially contribute to the development of the neurovascular response".

Critique: In Supplementary figure 3, it would nice to actually see the power frequency plots in either all of the different conditions or some key ones, like normocapnia, 10% CO₂ and acetylzolamide and even high current stim to see how neural activity has been boosted in this experiment. For the latter, it doesn't have to look identical to control but it is important to data to show.

Response: Thank you. We agree with the Reviewer that it is important to illustrate the profiles of the cortical neuronal responses to somatosensory stimulation. Supplementary Figure 3 had been revised accordingly and now includes the representative examples of neuronal spike power plotted over time for all five experimental conditions

Critique: There are two conflicting arguments in the new version of the paper: 1) all the blockers for known cell pathways in neurovascular coupling far from entirely block functional hyperemia, thus there is a significant unexplained pathway that could be CO₂; conflicts with: 2) meta-analysis of the literature reveals that common NVC blockers also block the effects of CO₂ reactivity; thus perhaps CO₂ is a driver of NVC through known pathways. Which stance would the authors like to take? Or can the wording be refined if these ideas are not mutually exclusive.

Response: We thank the Reviewer for making this observation and agree that the revision of our original submission resulted in two seemingly conflicting arguments in the new version of the paper. Our original hypothesis was that CO₂ actions mediate the proportion (30-40%) of the neurovascular coupling response which is insensitive to and remains after the experimental blockade of all hypothesized signaling pathways, suggested by the preceding studies. In response to the reviewers' comments raised during the first round of review, we conducted a systematic review and analysis of published data reporting the effects of pharmacological or genetic blockade of signaling mechanisms that were proposed to mediate the effects of CO₂ on the cerebral vasculature (Fig 1). Surprisingly, but in support of our main hypothesis, this analysis showed that CO₂ actions in the brain recruit the same vasodilatory signaling pathways, involving nitric oxide and cyclooxygenase products, that are implicated in mediating responses of cerebral blood vessels to the increases in the neuronal activity. Considering the outcome of this analysis of the literature data and

emerging new data on the potentially important role of K^+ signalling in driving the activity-dependent cerebrovascular dilations (see for example PMID:28319610), our modified hypotheses is that CO_2 produced by neurons is a key mediator of the neurovascular coupling response via the recruitment of signalling pathways involving nitric oxide and cyclooxygenase products. We now revised the text of the manuscript accordingly. It is important to note (and we make this point in the text of the paper) that while the vasodilatory signaling pathways that mediate the effects of CO_2 had been extensively researched, the cellular and molecular mechanisms underlying cerebrovascular CO_2 sensing remain unknown.

Critique: Cartoon showing Na^+ and HCO_3^- movement in and out of cells but grey circles are used for both species, which is confusing. Perhaps use different colours or draw a little stick molecule for HCO_3^- .

Response: We apologize that the labelling used in the cartoon illustrating CO_2 movements across the astroglial end-foot was not very clear. Grey circles depicted molecules of CO_2 ; Na^+ and HCO_3^- were not represented by the symbols. For clarity, we now removed Na^+ and HCO_3^- labels, but left the name of the transporter (NBCe1) as one of the conduits of CO_2 transport, as discussed above.

Critique: In the new figure 1, what does the triangle mean? The picture legend shows circles, but no triangles and I didn't see them defined in the legend text. Also, it would be helpful if the absolute size of the circles picture legend matched the absolute size of the circles in the figure. Right now, the smallest circle in the figure is much smaller than the smallest circle in the legend, which is a little confusing.

Response: We thank the reviewer for this comment and apologize for these imperfections. Triangles indicate studies conducted in human subjects, with the size and the color of the symbol representing the sample sizes. We now revised this Figure to make sure the absolute size of the circles in the figure legend matches the size of the circles in the figure. Thank you!

Reviewer #2

Thank you for taking the time to address my comments. I have the following minor comments regarding the revised version of the manuscript.

Critique: The addition of the meta-analysis is a nice touch and the remarks about concentration concerns and off target effects are reasonable, but the additional text is not clear to me - as I understand this you are referring to the data summarized in figure 2d and specifically to the means? This should be stated if so. Also, the 1.5 mA normocapnia condition is stated to be different but the P value of 0.07 does not indicate that there is a statistically significant difference between these two datasets. The text here should be adjusted to reflect this.

Response: We thank Dr Longden for his time taken to review our revised manuscript and apologize for the lack of clarity in our description of the effects of combined treatment with indomethacin and caffeine on the expression of the neurovascular response under different experimental conditions. For this revision we re-analyzed these data and the new analysis revealed that the effect of combined treatment with caffeine and indomethacin on the expression of the neurovascular response was only significant when forepaw stimulation

was applied with 3 mA current and in conditions of 10% inspired CO₂. We now revised the relevant text of the manuscript to read:

"As no significant CBF and BOLD responses to somatosensory stimulation were recorded at 10% inspired CO₂ following systemic treatment with caffeine/indomethacin (Figure 3c-e), this pharmacological treatment appeared to have an inhibitory effect on the neurovascular response in hypercapnia. CBF increases in the S1FL region induced by 3 mA stimulation at 10% inspired CO₂ were reduced by 29% (p=0.020) following systemic treatment with caffeine and indomethacin (Figure 3d). This result is consistent with the data reported in the literature⁵, suggesting that the signaling pathways sensitive to blockade by caffeine and indomethacin partially contribute to the development of the neurovascular response".

Reviewer #3

Critique: This paper posits that the signaling underlying neuromuscular coupling is poorly understood and that the role of CO₂ in the process had been unknown. I recall from Physiology lecture in the 1980s that CO₂ was the principle regulator of blood flow to the brain and underlies auto regulation. Moreover, the Bohr effect (1904) places CO₂ central in the gas exchange between hemoglobin and deoxyhemoglobin. Hence I and surely many others never questioned but assumed that CO₂ was the principle regulator of blood flow in the brain. I always considered the astrocytic regulation via PGE₂ and HETE an added mechanisms. I obviously stand corrected as this manuscript finally confirms what I had assumed to be the case.

Response: We thank this Referee for his time taken to review our revised manuscript but respectfully disagree with their assessment of the novelty of our work. Indeed, it is well known that cerebral vasculature is highly sensitive to CO₂ and changes in the arterial CO₂ have a major impact on cerebral blood flow. However, prior to our study the role of metabolically produced CO₂ in mediating the neuronal activity-dependent changes in local brain blood flow (neurovascular coupling response) had never been experimentally addressed. Moreover, the potential importance of metabolic feedback mechanisms (including increases in CO₂, decreases in O₂/glucose) in mediating the neurovascular response had been discussed and dismissed from the outset by many influential reviews on this topic (e.g. PMID:21068832).

Regarding the role of astrocytic regulation via PGE₂. The role of astroglial signalling in mediating the neurovascular coupling response had been challenged by evidence showing that mature astrocytes do not express appropriate glutamate receptors to sense the neuronal activity (PMID:23307741), neuronal activity-driven changes in local cerebral blood flow are not affected in the absence of astroglial IP₃ receptors (PMID:23658179; PMID:25253859; PMID:23785506) and only partially reduced when all hypothesized signalling pathways (including those mediated by PGE₂) are blocked pharmacologically (PMID:30481531). Moreover, the cerebral arteriole dilations often precede (PMID:23658179) and/or occur in the absence (PMID:26126870) of astrocytic Ca²⁺ responses.

Thus, there are still controversies surrounding the functional significance and the mechanisms underlying the neurovascular coupling response and the role of astrocytes. Our study provides the first experimental evidence in support of the hypothesis that CO₂ mediates the neurovascular coupling response by acting as a signaling molecule between

neurons and the cerebral vasculature and in our opinion contributes in a significant manner to our understanding of this fundamental mechanism.

There is an interesting recent publication studying the same question in humans: Caldwell HG, Howe CA, Hoiland RL, Carr JMJR, Chalifoux CJ, Brown CV, Patrician A, Tremblay JC, Panerai RB, Robinson TG, Minhas JS, Ainslie PN. Alterations in arterial CO₂ rather than pH affect the kinetics of neurovascular coupling in humans. *J Physiol*. 2021 Jun 9. doi: 10.1113/JP281615. Epub ahead of print. PMID: 34107079.

Response: We thank the Reviewer for drawing our attention to this publication. It was published while our paper was under review and supports the conclusions of our work. We now cite this study in the revised manuscript.

Critique: Of course in the present study using rodents the authors were able to examine the the astrocyte specific knockout of the Na bicarbonate exchanger showing that this is sufficient to block neuromuscular coupling. Yet rather than strengthening the conclusion this puzzles me as no mechanistic answer is provided as to how the astrocyte then causes the smooth muscle to relax. I could see how pH would do that but apparently not. So how does CO₂ cause vasodilation? Astrocytes seem to be essential yet how is not shown. Hence, while others may be enthusiastic about showing that CO₂ causes vasodilation and that this will supersede all other neuromuscular coupling, I find no new mechanistic insight in the paper.

Response: We agree that the question "how does CO₂ cause vasodilation?" – is important and relevant. For the first revision of our paper, we reviewed hundreds of publications and carried out a systematic review and analysis of published data reporting the effects of pharmacological or genetic blockade of all hypothesized mechanisms that were proposed to mediate the effects of CO₂ on cerebral vasculature (Fig 1). While the vasodilatory signaling pathways that mediate the CO₂ effects had been extensively researched, the cellular and molecular mechanisms underlying cerebrovascular CO₂ *sensing* remain unknown. Considering significant research effort by many research groups over the last 20 years, identification of these mechanisms is not a trivial task, and this work is ongoing. Our data point to a critical importance of NBCe1 expressed in astrocytes, which, as we hypothesize, is functioning as an important conduit of CO₂/HCO₃⁻ transport in the brain.

We respectfully disagree with the reviewer that our work has no new mechanistic insight. In our manuscript we present several lines of experimental evidence suggesting that neuronal activity-dependent changes in local cerebral blood flow are mediated by CO₂. As we discuss in our manuscript and argue above, this represents an important conceptual advance in our understanding of the mechanisms of neurovascular coupling which is expected to have a significant impact in the field.

Critique: I am a bit puzzled by this statement: "that CO₂ diffusion across cellular membranes is facilitated by certain membrane channels and transporters" ...gases move freely across membranes irrespective of channels or transporters. also it is misleading to refer to "CO₂/HCO₃⁻ transport in astrocytes" as CO₂ is not transported. The transporter exchanges Na and bicarbonate.

Response: Indeed, historically CO₂ was thought to diffuse through the plasma membrane freely. A significant body of work by Walter Boron and colleagues has led to the understanding that the diffusion of CO₂ across the biological membrane is restricted and to a large extent is mediated by certain membrane channels, such as aquaporin water channels. More recent evidence suggests that CO₂ can also diffuse through the NBCe1. In a

recent review article Michenkova and colleagues (PMID: 33633837) discuss the results of three different assays that all provided evidence in support of the hypothesis that NBCe1, in the presence of HCO_3^- , has a substantial CO_2 conductance. The authors suggested that either bicarbonate has an allosteric effect on NBCe1, thereby creating a CO_2 channel through the protein and/or NBCe1 cycles through conformational changes during $\text{Na}^+/\text{HCO}_3^-$ cotransport with CO_2 moving through a transiently available CO_2 pathway. Characteristically high expression of NBCe1 in astrocytes can, therefore, be explained by this property of the transporter which together with aquaporin channels are responsible for the effective CO_2 transport across the astroglial membrane.

REVIEWER COMMENTS

Reviewer #1 (Remarks to the Author):

The authors have satisfactorily addressed my comments. Nothing further required.

Reviewer #2 (Remarks to the Author):

Thank you very much for addressing my concerns.

Reviewer #4 (Remarks to the Author):

Hosford and colleagues investigate how the vasodilatory actions of CO₂ interacts with neurovascular coupling. The authors use BOLD/CBF measures in rats and use NBCe KO mice to investigate how altering CO₂ levels independent of pH/baseline flow occlude the response to electrical stimulation of the paw. The authors find that elevated CO₂ levels, and not just lowered pH, has a vasodilatory effect and that this can occlude normal neurovascular coupling. From this and a meta-analysis study, the authors conclude that neurovascular coupling and CO₂ act via common pathways (potentially cxn26 and other connexins). I have concerns about the quality of the data and the strength of the conclusions that can be drawn from them.

Major issues

the fMRI has too many confounds to measure neurovascular coupling reliably under the conditions used in this study (where baseline blood volume/flow changes drastically). These measures are low signal to noise, and the flow changes may be mediated by dilation/constriction of large arteries outside the ROI. The authors need to perform some direct measures of vascular dynamics (like 2-photon microscopy) with single vessel resolution to show that the CO₂ is acting at the same site as traditional neurovascular coupling.

To study neurovascular coupling, one needs to make measures of neural activity. The reduced CBF response during hypercapnia could simply be caused by a reduction in evoked neural activity, but the authors measurement techniques are too crude to determine this. The authors used average absolute value of evoked potentials, which is used in some neurovascular coupling experiments, but not in high quality research in other fields. This measure is not clean, and is not something that electrophysiologist have relied upon in the last half-century. The amplitude of the evoked potential will depend more on the level of background activity and how synchronized the neural activity is than the absolute increase in neural firing. The evoked potential and firing rates may be differentially affected by the drugs and perturbations used here, making comparisons across conditions difficult. More standard measures of neural activity (using multiunit spiking, LFP power, or GCamp signals) have shown clear, quantitative correspondence between neural activity and vascular responses (Cardoso et al., Nature Neuro 2015; Ma et al, PNAS 2016). In vitro studies have shown that elevated CO₂ greatly decreases the excitability of neurons (Dulla et al., 2005, Neuron). There is likely a non-linear synergy between the depressant effects of the anesthetic and CO₂. Given the data presented here, I don't think the hypothesis that high levels of CO₂ inhibit neural activity, and thus an occlusion of NVC in this manner can be ruled out by the experiments here.

All of the available human data that I am aware of contradicts the authors hypothesis and observations. In humans, hypercapnia raises baseline CBF, but does not alter the absolute magnitude of the stimulus evoked CBF, or if so, only slightly (Kastrup et al., Neuroreport 1999; Cohen et al., JCBFM; Whittaker et al., 2016, Neuroimage), supporting an additive interaction between the CO₂ and NVC pathways, not a common mechanism. Similar additive interactions between CO₂ and NVC have been found in animals (Jones et al., 2005, Neuroimage; Nishino et al., Microcirculation 2015). This supports the idea that CO₂ acts independently of other neurally released vasodilators. While there may be species differences, the human data is in non-anesthetized subjects (unlike the study here),

and due to the larger size of the brain the signal to noise should be higher. As discussed above, CO₂ inhibits neural activity, and this hypercapnia-mediated inhibition of neural activity combined with the anesthesia used here might explain the difference.

Even disregarding all these other issues, I think the conclusion that CO₂ mediates NVC is unfounded. There is no evidence presented that the amount of CO₂ produced by neural activity is capable of dilating blood vessels, and that it happens on a fast enough timescale to underly NVC (dilation within ~1 second of stimulus onset). Measurements of brain pH during stimulation show an initial *alkaline* transient following neural activity, inconsistent with the authors hypothesis (reviewed in Chesler 2003, Physiological rev.). If CO₂ levels rose in response to metabolic activity, there should be an acidic transient that should track the CO₂ levels.

There is a lack of positive controls that NBCe1 transporter knockdown changes extracellular pH, and negative control that neural activity is not elevated in these mice. Oxygen signals are not good measures of blood flow/vasodilation change (they are affected by metabolism and don't always reflect the vascular response), the authors should use laser Doppler or optical imaging to assay the mice.

Minor

In 2e-g, bottom is labeled "acetazolamide", should be "normocapnia" and in blue for consistency. The number of animals used in each panel should be labeled either in the figure or the legend. The color scaling on the raw CBF images is poor, rendering them uninterpretable. The authors should use a log scale in order to capture the very wide range of flows.

Manuscript ID: NCOMMS-20-46461B
Responses to the referees' comments

We would like to thank the reviewers and the Editors for their time taken to evaluate our revised submission. We are pleased that the original reviewers of our paper gave a very positive assessment of our work (Reviewer 1: "I think this study is important, novel and tackling an age-old question") and are fully satisfied with our responses to their comments and our extensive revisions of the manuscript. We are grateful for further comments provided by an additional reviewer (Reviewer 4). Reviewer 4 provided comments on several aspects of our work that were not in line with the recommendation given by the other three reviewers. Here we provide a full response to all the criticisms raised by Reviewer 4 and revised the text and figures of the manuscript as appropriate.

Reviewer #4:

Hosford and colleagues investigate how the vasodilatory actions of CO₂ interacts with neurovascular coupling. The authors use BOLD/CBF measures in rats and use NBCe KO mice to investigate how altering CO₂ levels independent of pH/baseline flow occlude the response to electrical stimulation of the paw. The authors find that elevated CO₂ levels, and not just lowered pH, has a vasodilatory effect and that this can occlude normal neurovascular coupling. From this and a meta-analysis study, the authors conclude that neurovascular coupling and CO₂ act via common pathways (potentially cxn26 and other connexins). I have concerns about the quality of the data and the strength of the conclusions that can be drawn from them.

Response: We thank this referee for his/her time taken to review our paper, but respectfully disagree with their overall assessment of our work. We are puzzled by the ambiguity of some of the comments raised and provide our detailed responses below.

Critique: the fMRI has too many confounds to measure neurovascular coupling reliably under the conditions used in this study (where baseline blood volume/flow changes drastically).

Response: Thank you for this comment. In all the experiments described in the paper we took into the account and carefully controlled for the potential changes in all the important variables. The answer to this comment of the reviewer is provided in the text of the manuscript: "We implemented an ASL sequence with T2* weighted imaging for combined measurement of local CBF (in ml 100 g⁻¹ min⁻¹) and BOLD signal changes in the S1FL region of the cortex. The measurements of absolute changes in CBF were essential to address the objectives of this study as the BOLD signal reflects an unknown combination of changes in local CBF, blood volume and CMRO₂. Arterial spin labelling (ASL) measurements provided absolute values of cerebral perfusion to control for possible confounding effects of the experimental manipulations on the expression of the neurovascular response".

In this study we carefully designed and implemented an experimental paradigm that uses ASL imaging to reliably capture spatially specific increases in CBF due to neurovascular coupling across different baseline blood flow conditions. This is demonstrated, for example, in Figure 2C where a clear marked local CBF increase is recorded due to the neurovascular coupling despite elevated baseline global CBF induced by systemic acetazolamide treatment.

Critique: These measures are low signal to noise, and the flow changes may be mediated by dilation/constriction of large arteries outside the ROI.

Response: Respectfully we disagree with the reviewer here. We reliably capture robust CBF responses to increases in the local neuronal activity (see Figure 2C) using a forepaw stimulation paradigm that we have previously optimized and characterized (e.g. PMID: 25834053; PMID: 28079057). In control conditions (in the absence of supplemental CO₂/drugs), we see specific increases in CBF in the contralateral (to the stimulated paw/limb) S1FL region. This is illustrated in the examples of ASL perfusion weighted (control-label) images below showing that specific increases in CBF are only detected in the contralateral somatosensory cortex.

These spatially restricted signals cannot be possibly explained by dilation/constriction of large arteries outside the ROI, as these would become obvious in whole brain imaging. In the revised manuscript we now provide these images and include additional data showing changes in CBF induced by forepaw stimulation within the ipsilateral S1FL region (revised Supplementary Figure 1B). The regional specificity of the CBF response is illustrated on a group level where the equivalent CBF time-course data are plotted to that presented in Figure 2C, but for ROIs in the ipsilateral somatosensory cortex.

Critique: The authors need to perform some direct measures of vascular dynamics (like 2-photon microscopy) with single vessel resolution to show that the CO₂ is acting at the same site as traditional neurovascular coupling.

Response: Respectfully we do not understand what this Reviewer means by 'traditional neurovascular coupling'. Neurovascular coupling describes increases in local cerebral blood flow in response to increases in the neuronal activity. As discussed above, ASL fMRI can reliably capture changes in local CBF and, therefore, represents one of the most robust methods that can be used to study the mechanisms underlying the neurovascular coupling response. 2-photon imaging with single vessel resolution provides data on vascular dynamics in a small brain area, and (as reviewer points above) in this case the data obtained can indeed be confounded by dilation/constriction of large arteries outside this area, but in conditions when these potential responses are not monitored and/or controlled for.

Critique: To study neurovascular coupling, one needs to make measures of neural activity. The reduced CBF response during hypercapnia could simply be caused by a reduction in evoked neural activity, but the authors measurement techniques are too crude to determine this. The authors used average absolute value of evoked potentials, which is used in some neurovascular coupling experiments, but not in high quality research in other fields. This measure is not clean, and is not something that electrophysiologist have relied upon in the last half-century. The amplitude of the evoked potential will depend more on the level of background activity and how synchronized the neural activity is than the absolute increase in neural firing. The evoked potential and firing rates may be differentially affected by the drugs and perturbations used here, making comparisons across conditions difficult. More

standard measures of neural activity (using multiunit spiking, LFP power, or GCamp signals) have shown clear, quantitative correspondence between neural activity and vascular responses (Cardoso et al., Nature Neuro 2015; Ma et al, PNAS 2016). In vitro studies have shown that elevated CO₂ greatly decreases the excitability of neurons (Dulla et al., 2005, Neuron). There is likely a non-linear synergy between the depressant effects of the anesthetic and CO₂. Given the data presented here, I don't think the hypothesis that high levels of CO₂ inhibit neural activity, and thus an occlusion of NVC in this manner can be ruled out by the experiments here.

Response: Respectfully we disagree with the reviewer here. Recordings of the neuronal activity were performed in all the experimental conditions to make sure that the effects of drugs and experimental treatments on cortical neuronal excitability are recorded and controlled for. As we discuss in the paper this was done to ensure we make appropriate comparisons between different experimental conditions. All the data are illustrated, described, and discussed in detail. As expected, the inhibitory effect of CO₂ was observed, and stimulation parameters were adjusted to fully compensate for CO₂-induced neuronal inhibition. Recordings using carbon fiber microelectrodes give accurate measures of evoked neuronal activity in the targeted area of the brain. The reviewer does not provide any argument in support of their view that the recordings using carbon fibre microelectrodes are "crude" and "not clean". It should be made clear that the background activity was subtracted from the recordings during each condition to isolate the evoked activity and allow direct comparison between neuronal activity recorded under different experimental conditions. The carbon fiber microelectrode recordings in this study are of Local Field Potentials, but the lower impedance of the carbon fibre may produce an extracellular potential shape that is unfamiliar. Below we provide examples of the evoked neuronal activity recorded in the rat and mouse S1FL region in response to a pulse of electrical stimulation of the forepaw. We agree that perhaps the raw recordings were not well illustrated in the previous version of the paper and now revised Figure 4 to include an example of the evoked potentials recorded in the S1FL region of a mouse.

The profiles of the evoked neuronal activity in the S1FL region in response to forepaw stimulation recorded in our study were very similar to that reported in a classical study by Logothetis and colleagues involving simultaneous recordings of local field potentials/multiunit neuronal activity and BOLD signals (PMID: 11449264). We were also surprised by the Reviewer's view that GCamp signals are standard measures of neuronal activity; genetically-encoded Ca²⁺ indicators are not at all accurate in reporting neuronal inhibition, especially when the baseline activity is low.

Critique: All of the available human data that I am aware of contradicts the authors hypothesis and observations. In humans, hypercapnia raises baseline CBF, but does not alter the absolute magnitude of the stimulus evoked CBF, or if so, only slightly (Kastrup et al., Neuroreport 1999; Cohen et al., JCBFM; Whittaker et al., 2016, Neuroimage), supporting an additive interaction between the CO₂ and NVC pathways, not a common

mechanism. Similar additive interactions between CO₂ and NVC have been found in animals (Jones et al., 2005, Neuroimage; Nishino et al., Microcirculation 2015). This supports the idea that CO₂ acts independently of other neurally released vasodilators. While there may be species differences, the human data is in non-anesthetized subjects (unlike the study here), and due to the larger size of the brain the signal to noise should be higher. As discussed above, CO₂ inhibits neural activity, and this hypercapnia-mediated inhibition of neural activity combined with the anesthesia used here might explain the difference.

Response: We thank the reviewer for raising this important comment. We are aware of all the preceding literature:

In the Whittaker et al. (2016) study, 5% CO₂ was added to the inspired air to increase P_{ET} CO₂ by 6 mmHg. This treatment increased basal CBF by ~50% but had no effect on the expression of the neurovascular response to visual stimuli.

In the Cohen et al (2002) study, 5% CO₂ was given in the inspired air; P_{ET}CO₂ increased by ~10 mmHg on average. CBF data were not obtained/reported, but it was a significant (by ~30%) reduction of the peak of visually-evoked BOLD response.

Jones et al. (2005) did not report “additive interactions between CO₂ and NVC”, but the data presented in that paper show a significant reduction of the CBF response and almost complete blockade of the estimated BOLD response in the rat barrel cortex induced by whisker stimulation in conditions of 10% inspired CO₂.

There is also no data on “additive interactions between CO₂ and NVC” in the Nishino et al. (2015) study. In this investigation conducted in 10 mice 5% CO₂ was given in the inspired air and whiskers were stimulated by an air puff. Figure 6 of that paper shows that the percentage increase in the vessel diameter induced by 5% CO₂ + sensory stimulation is much lower compared to the sum of the responses induced by these two stimuli given separately. This study reported a very small (~10%) increase in CBF in response to 5% inspired CO₂ which is inconsistent with the other data reported in the literature.

In our opinion the data reported in the publications mentioned by the reviewer are in full agreement with the results of our experiments. We show a partial inhibition of the neurovascular response when 5% CO₂ is given in the inspired air and a marked reduction of the response in conditions of 10% inspired CO₂. Assuming that within a physiological range there is a linear relationship between brain tissue PCO₂ and cerebral blood flow, we may reasonably expect activity-dependent increases in CBF (driven by CO₂ produced as a result of increased neuronal activity) developing on top of the baseline, elevated by inspired CO₂. Thus, the argument of the Reviewer about the “additive” interactions between CO₂ and neurovascular signalling cannot be used to dismiss our hypothesis. Much higher levels of exogenous CO₂ (compared to what can be used in human studies) are required to saturate the CO₂-sensitive vasodilatory mechanism and occlude the actions of CO₂ produced by neurons. The experiments of this type can only be done in anaesthetized animals, but not in non-anesthetized human subjects. We now include a brief discussion of this point in the revised manuscript. Thank you.

Critique: Even disregarding all these other issues, I think the conclusion that CO₂ mediates NVC is unfounded. There is no evidence presented that the amount of CO₂ produced by neural activity is capable of dilating blood vessels, and that it happens on a fast enough timescale to underly NVC (dilation within ~1 second of stimulus onset).

Response: Respectfully we disagree that this reasoning can be used to disregard the hypothesis that CO₂ mediates the neurovascular coupling response. The human brain generates ~20% of total body CO₂ production; most of glucose metabolised by the brain tissue ends up as CO₂ in neurons. Thus, there is plenty of CO₂ produced by active neurons to modulate cerebral blood flow. Brain CO₂ production is proportional to the neuronal energy use. Only small changes in CO₂ concentration are required to gate certain CO₂ sensitive channels (PMID: 24220509) that can act as conduits of release of vasoactive signalling molecules. CO₂ is a fast-diffusing molecule, and its rate of diffusion is unlikely to be slower compared to the other molecules implicated in the development of the neurovascular coupling response (such as glutamate).

Critique: Measurements of brain pH during stimulation show an initial *alkaline* transient following neural activity, inconsistent with the authors hypothesis (reviewed in Chesler 2003, Physiological rev.). If CO₂ levels rose in response to metabolic activity, there should be an acidic transient that should track the CO₂ levels.

Response: Aerobic respiration generates CO₂, therefore, CO₂ is always produced as a result of metabolic activity. The *alkaline* transient associated with the increases in neuronal activity is due to rapid release of bicarbonate by astrocytes via the mechanism we recently described (PMID: 33033238). CO₂ may have a direct action on certain membrane channels (PMID: 24220509), independently of pH changes. In his review, Chesler points out that alkalinization is not an ubiquitous response to neuronal activity in every brain area, however the only arguably ubiquitous response to an increase in neuronal activity is the production of CO₂ due to the necessary coupled increase in the metabolic rate. Thus, the observations of a well-maintained extracellular pH or an *alkaline* transient following neuronal activity cannot be used as arguments to suggest that CO₂ is not produced in response to metabolic activity and/or has no signalling role on its own.

Critique: There is a lack of positive controls that NBCe1 transporter knockdown changes extracellular pH, and negative control that neural activity is not elevated in these mice.

Response: In our recent publication we reported validation of this animal model and demonstrated that NBCe1 knockdown in cortical astrocytes impairs control of extracellular pH (PMID: 33033238). Neural activity was recorded, and it was shown not to be different between the experimental groups.

Critique: Oxygen signals are not good measures of blood flow/vasodilation change (they are affected by metabolism and don't always reflect the vascular response), the authors should use laser Doppler or optical imaging to assay the mice.

Response: In our recent study we demonstrated a very good correlation between the recordings of the neurovascular coupling response obtained using oxygen measurements with carbon fiber microelectrodes and BOLD signals obtained using fMRI scanner (PMID: 32685919), but we agree that laser Doppler may provide additional information.

Minor:

In 2e-g, bottom is labeled "acetazolamide", should be "normocapnia" and in blue for consistency.

Now corrected. Thank you.

The number of animals used in each panel should be labeled either in the figure or the legend.

Now indicated. Thank you.

The color scaling on the raw CBF images is poor, rendering them uninterpretable. The authors should use a log scale in order to capture the very wide range of flows.

Thank you. We now improved the color scaling of the raw CBF images shown on Fig 3b.

REVIEWER COMMENTS

Reviewer #4 (Remarks to the Author):

no further comments